# VAEBM: A Symbiosis between Variational Autoencoders and Energy-based Models

**Zhisheng Xiao** [*]
Computational and Applied Mathematics
The University of Chicago
zxiao@uchicago.edu

**Karsten Kreis, Jan Kautz, Arash Vahdat**
NVIDIA
{kkreis,jkautz,avahdat}@nvidia.com

## ABSTRACT

Energy-based models (EBMs) have recently been successful in representing complex distributions of small images. However, sampling from them requires expensive Markov chain Monte Carlo (MCMC) iterations that mix slowly in high dimensional pixel space. Unlike EBMs, variational autoencoders (VAEs) generate samples quickly and are equipped with a latent space that enables fast traversal of the data manifold. However, VAEs tend to assign high probability density to regions in data space outside the actual data distribution and often fail at generating sharp images. In this paper, we propose VAEBM, a symbiotic composition of a VAE and an EBM that offers the best of both worlds. VAEBM captures the overall mode structure of the data distribution using a state-of-the-art VAE and it relies on its EBM component to explicitly exclude non-data-like regions from the model and refine the image samples. Moreover, the VAE component in VAEBM allows us to speed up MCMC updates by reparameterizing them in the VAE's latent space. Our experimental results show that VAEBM outperforms state-of-the-art VAEs and EBMs in generative quality on several benchmark image datasets by a large margin. It can generate high-quality images as large as $256 \times 256$ pixels with short MCMC chains. We also demonstrate that VAEBM provides complete mode coverage and performs well in out-of-distribution detection.

## 1 INTRODUCTION

Deep generative learning is a central problem in machine learning. It has found diverse applications, ranging from image (Brock et al., 2018; Karras et al., 2019; Razavi et al., 2019), music (Dhariwal et al., 2020) and speech (Ping et al., 2020; Oord et al., 2016a) generation, distribution alignment across domains (Zhu et al., 2017; Liu et al., 2017; Tzeng et al., 2017) and semi-supervised learning (Kingma et al., 2014; Izmailov et al., 2020) to 3D point cloud generation (Yang et al., 2019), light-transport simulation (Müller et al., 2019), molecular modeling (Sanchez-Lengeling & Aspuru-Guzik, 2018; Noé et al., 2019) and equivariant sampling in theoretical physics (Kanwar et al., 2020).

Among competing frameworks, likelihood-based models include variational autoencoders (VAEs) (Kingma & Welling, 2014; Rezende et al., 2014), normalizing flows (Rezende & Mohamed, 2015; Dinh et al., 2016), autoregressive models (Oord et al., 2016b), and energy-based models (EBMs) (Lecun et al., 2006; Salakhutdinov et al., 2007). These models are trained by maximizing the data likelihood under the model, and unlike generative adversarial networks (GANs) (Goodfellow et al., 2014), their training is usually stable and they cover modes in data more faithfully by construction.

Among likelihood-based models, EBMs model the unnormalized data density by assigning low energy to high-probability regions in the data space (Xie et al., 2016; Du & Mordatch, 2019). EBMs are appealing because they require almost no restrictions on network architectures (unlike normalizing flows) and are therefore potentially very expressive. They also exhibit better robustness and out-of-distribution generalization (Du & Mordatch, 2019) because, during training, areas with high probability under the model but low probability under the data distribution are penalized explicitly. However, training and sampling EBMs usually requires MCMC, which can suffer from slow mode mixing and is computationally expensive when neural networks represent the energy function.

---

[*]Work done during an internship at NVIDIA

On the other hand, VAEs are computationally more efficient for sampling than EBMs, as they do not require running expensive MCMC steps. VAEs also do not suffer from expressivity limitations that normalizing flows face (Dupont et al., 2019; Kong & Chaudhuri, 2020), and in fact, they have recently shown state-of-the-art generative results among non-autoregressive likelihood-based models (Vahdat & Kautz, 2020). Moreover, VAEs naturally come with a latent embedding of data that allows fast traverse of the data manifold by moving in the latent space and mapping the movements to the data space. However, VAEs tend to assign high probability to regions with low density under the data distribution. This often results in blurry or corrupted samples generated by VAEs. This also explains why VAEs often fail at out-of-distribution detection (Nalisnick et al., 2019).

In this paper, we propose a novel generative model as a symbiotic composition of a VAE and an EBM (VAEBM) that combines the best of both. VAEBM defines the generative distribution as the product of a VAE generator and an EBM component defined in pixel space. Intuitively, the VAE captures the majority of the mode structure in the data distribution. However, it may still generate samples from low-probability regions in the data space. Thus, the energy function focuses on refining the details and reducing the likelihood of non-data-like regions, which leads to significantly improved samples.

Moreover, we show that training VAEBM by maximizing the data likelihood easily decomposes into training the VAE and the EBM component separately. The VAE is trained using the reparameterization trick, while the EBM component requires sampling from the joint energy-based model during training. We show that we can sidestep the difficulties of sampling from VAEBM, by reparametrizing the MCMC updates using VAE's latent variables. This allows MCMC chains to quickly traverse the model distribution and it speeds up mixing. As a result, we only need to run short chains to obtain approximate samples from the model, accelerating both training and sampling at test time.

Experimental results show that our model outperforms previous EBMs and state-of-the-art VAEs on image generation benchmarks including CIFAR-10, CelebA 64, LSUN Church 64, and CelebA HQ 256 by a large margin, reducing the gap with GANs. We also show that our model covers the modes in the data distribution faithfully, while having less spurious modes for out-of-distribution data. To the best of knowledge, VAEBM is the first successful EBM applied to large images.

In summary, this paper makes the following contributions: i) We propose a new generative model using the product of a VAE generator and an EBM defined in the data space. ii) We show how training this model can be decomposed into training the VAE first, and then training the EBM component. iii) We show how MCMC sampling from VAEBM can be pushed to the VAE's latent space, accelerating sampling. iv) We demonstrate state-of-the-art image synthesis quality among likelihood-based models, confirm complete mode coverage, and show strong out-of-distribution detection performance.

## 2 BACKGROUND

**Energy-based Models:** An EBM assumes $p_\psi(\mathbf{x})$ to be a Gibbs distribution of the form $p_\psi(\mathbf{x}) = \exp\left(-E_\psi(\mathbf{x})\right)/Z_\psi$, where $E_\psi(\mathbf{x})$ is the energy function with parameters $\psi$ and $Z_\psi = \int_\mathbf{x} \exp\left(-E_\psi(\mathbf{x})\right) d\mathbf{x}$ is the normalization constant. There is no restriction on the particular form of $E_\psi(\mathbf{x})$. Given a set of samples drawn from the data distribution $p_d(\mathbf{x})$, the goal of maximum likelihood learning is to maximize the log-likelihood $L(\psi) = \mathbb{E}_{\mathbf{x} \sim p_d(\mathbf{x})}\left[\log p_\psi(\mathbf{x})\right]$, which has the derivative (Woodford, 2006):

$$\partial_\psi L(\psi) = \mathbb{E}_{x \sim p_d(\mathbf{x})}\left[-\partial_\psi E_\psi\left(\mathbf{x}\right)\right] + \mathbb{E}_{x \sim p_\psi(\mathbf{x})}\left[\partial_\psi E_\psi\left(\mathbf{x}\right)\right] \tag{1}$$

For the first expectation, the *positive phase*, samples are drawn from the data distribution $p_d(\mathbf{x})$, and for the second expectation, the *negative phase*, samples are drawn from the model $p_\psi(\mathbf{x})$ itself. However, sampling from $p_\psi(\mathbf{x})$ in the negative phase is itself intractable and approximate samples are usually drawn using MCMC. A commonly used MCMC algorithm is Langevin dynamics (LD) (Neal, 1993). Given an initial sample $\mathbf{x}_0$, Langevin dynamics iteratively updates it as:

$$\mathbf{x}_{t+1} = \mathbf{x}_t - \frac{\eta}{2}\nabla_\mathbf{x} E_\psi(\mathbf{x}_t) + \sqrt{\eta}\omega_t, \quad \omega_t \sim \mathcal{N}(0, \mathbf{I}), \tag{2}$$

where $\eta$ is the step-size.[1] In practice, Eq. 2 is run for finite iterations, which yields a Markov chain with an invariant distribution approximately close to the original target distribution.

---

[1] In principle one would require an accept/reject step to make it a rigorous MCMC algorithm, but for sufficiently small stepsizes this is not necessary in practice (Neal, 1993).

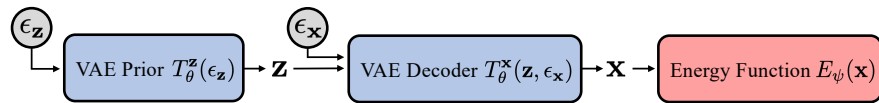

Figure 1: Our VAEBM is composed of a VAE generator (including the prior and decoder) and an energy function that operates on samples $\mathbf{x}$ generated by the VAE. The VAE component is trained first, using the standard VAE objective; then, the energy function is trained while the generator is fixed. Using the VAE generator, we can express the data variable $\mathbf{x}$ as a deterministic function of white noise samples $\boldsymbol{\epsilon_z}$ and $\boldsymbol{\epsilon_x}$. This allows us to reparameterize sampling from our VAEBM by sampling in the joint space of $\boldsymbol{\epsilon_z}$ and $\boldsymbol{\epsilon_x}$. We use this in the negative training phase (see Sec. 3.1).

**Variational Autoencoders:** VAEs define a generative model of the form $p_\theta(\mathbf{x}, \mathbf{z}) = p_\theta(\mathbf{z})p_\theta(\mathbf{x}|\mathbf{z})$, where $\mathbf{z}$ is the latent variable with prior $p_\theta(\mathbf{z})$, and $p_\theta(\mathbf{x}|\mathbf{z})$ is a conditional distribution that models the likelihood of data $\mathbf{x}$ given $\mathbf{z}$. The goal of training is to maximize the marginal log-likelihood $\log p_\theta(\mathbf{x})$ given a set of training examples. However since the marginalization is intractable, instead, the variational lower bound on $\log p_\theta(\mathbf{x})$ is maximized with $q_\phi(\mathbf{z}|\mathbf{x})$ as the approximate posterior:

$$\log p_\theta(\mathbf{x}) \geq \mathbb{E}_{\mathbf{z} \sim q_\phi(\mathbf{z}|\mathbf{x})}\left[\log p_\theta(\mathbf{x}|\mathbf{z})\right] - D_{\mathrm{KL}}\left[q_\phi(\mathbf{z}|\mathbf{x})\|p_\theta(\mathbf{z})\right] := \mathcal{L}_{\mathrm{vae}}(\mathbf{x}, \theta, \phi). \quad (3)$$

The state-of-the-art VAE, NVAE (Vahdat & Kautz, 2020), increases the expressivity of both prior and approximate posterior using hierarchical latent variables (Kingma et al., 2016) where $\mathbf{z}$ is decomposed into a set of disjoint groups, $\mathbf{z} = \{\mathbf{z}_1, \mathbf{z}_1, \ldots, \mathbf{z}_L\}$, and the prior $p_\theta(\mathbf{z}) = \prod_l p_\theta(\mathbf{z}_l|\mathbf{z}_{<l})$ and the approximate posterior $q_\phi(\mathbf{z}|\mathbf{x}) = \prod_l q_\phi(\mathbf{z}_l|\mathbf{z}_{<l}, \mathbf{x})$ are defined using autoregressive distributions over the groups. We refer readers to Vahdat & Kautz (2020) for more details.

## 3 ENERGY-BASED VARIATIONAL AUTOENCODERS

One of the main problems of VAEs is that they tend to assign high probability to regions in data space that have low probability under the data distribution. To tackle this issue, we propose VAEBM, a generative model constructed by the product of a VAE generator and an EBM component defined in the data space. This formulation allows our model to capture the main mode structure of the data distribution using the VAE. But when training the joint VAEBM, in the negative training phase we sample from the model itself and can discover non-data-like samples, whose likelihood is then reduced by the energy function explicitly. The energy function defined in the pixel space also shares similarities with discriminator in GANs, which can generate crisp and detailed images.

Formally, we define the generative model in VAEBM as $h_{\psi,\theta}(\mathbf{x}, \mathbf{z}) = \frac{1}{Z_{\psi,\theta}}p_\theta(\mathbf{x}, \mathbf{z})e^{-E_\psi(\mathbf{x})}$ where $p_\theta(\mathbf{x}, \mathbf{z}) = p_\theta(\mathbf{z})p_\theta(\mathbf{x}|\mathbf{z})$ is a VAE generator and $E_\psi(\mathbf{x})$ is a neural network-based energy function, operating only in the $\mathbf{x}$ space, and $Z_{\psi,\theta} = \int p_\theta(\mathbf{x})e^{-E_\psi(\mathbf{x})}d\mathbf{x}$ is the normalization constant. VAEBM is visualized in Fig. 1. Marginalizing out the latent variable $\mathbf{z}$ gives

$$h_{\psi,\theta}(\mathbf{x}) = \frac{1}{Z_{\psi,\theta}}\int p_\theta(\mathbf{x}, \mathbf{z})e^{-E_\psi(\mathbf{x})}d\mathbf{z} = \frac{1}{Z_{\psi,\theta}}p_\theta(\mathbf{x})e^{-E_\psi(\mathbf{x})}. \quad (4)$$

Given a training dataset, the parameters of VAEBM, $\psi, \theta$, are trained by maximizing the marginal log-likelihood on the training data:

$$\log h_{\psi,\theta}(\mathbf{x}) = \log p_\theta(\mathbf{x}) - E_\psi(\mathbf{x}) - \log Z_{\psi,\theta} \quad (5)$$

$$\geq \underbrace{\mathbb{E}_{\mathbf{z} \sim q_\phi(\mathbf{z}|\mathbf{x})}[\log p_\theta(\mathbf{x}|\mathbf{z}) - D_{\mathrm{KL}}(q_\phi(\mathbf{z}|\mathbf{x})\|p(\mathbf{z}))}_{\mathcal{L}_{\mathrm{vae}}(\mathbf{x}, \theta, \phi)} \underbrace{-E_\psi(\mathbf{x}) - \log Z_{\psi,\theta}}_{\mathcal{L}_{\mathrm{EBM}}(\mathbf{x}, \psi, \theta)}, \quad (6)$$

where we replace $\log p_\theta(\mathbf{x})$ with the variational lower bound from Eq. 3. Eq. 6 forms the objective function for training VAEBM. The first term corresponds to the VAE objective and the second term corresponds to training the EBM component. Next, we discuss how we can optimize this objective.

## 3.1 TRAINING

The $\mathcal{L}_{\text{EBM}}(\mathbf{x}, \psi, \theta)$ term in Eq. 6 is similar to the EBM training objective except that the log partition function depends on both $\psi$ and $\theta$. We show in Appendix A that $\log Z_{\psi,\theta}$ has the gradients

$$\partial_\psi \log Z_{\psi,\theta} = \mathbb{E}_{\mathbf{x} \sim h_{\psi,\theta}(\mathbf{x},\mathbf{z})} \left[ -\partial_\psi E_\psi(\mathbf{x}) \right] \quad \text{and} \quad \partial_\theta \log Z_{\psi,\theta} = \mathbb{E}_{\mathbf{x} \sim h_{\psi,\theta}(\mathbf{x},\mathbf{z})} \left[ \partial_\theta \log p_\theta(\mathbf{x}) \right].$$

The first gradient can be estimated easily by evaluating the gradient of the energy function at samples drawn from the VAEBM model $h_{\psi,\theta}(\mathbf{x}, \mathbf{z})$ using MCMC. However, the second term involves computing the intractable $\frac{\partial}{\partial \theta} \log p_\theta(\mathbf{x})$. In Appendix A, we show that estimating $\frac{\partial}{\partial \theta} \log p_\theta(\mathbf{x})$ requires sampling from the VAE's posterior distribution, given model samples $\mathbf{x} \sim h_{\psi,\theta}(\mathbf{x}, \mathbf{z})$. To avoid the computational complexity of estimating this term, for example with a second round of MCMC, we propose a two-stage algorithm for training VAEBM. In the first stage, we train the VAE model in our VAEBM by maximizing the $\mathcal{L}_{\text{vae}}(\mathbf{x}, \theta, \phi)$ term in Eq. 6. This term is identical to the VAE's objective, thus, the parameters $\theta$ and $\phi$ are trained using the reparameterized trick as in Sec. 2. In the second stage, we keep the VAE model fixed and only train the EBM component. Since $\theta$ is now fixed, we only require optimizing $\mathcal{L}_{\text{EBM}}(\mathbf{x}, \psi, \theta)$ w.r.t. $\psi$, the parameters of the energy function. The gradient of $L(\psi) = \mathbb{E}_{\mathbf{x} \sim p_d} \left[ \mathcal{L}_{\text{EBM}}(\mathbf{x}, \psi, \theta) \right]$ w.r.t. $\psi$ is:

$$\partial_\psi L(\psi) = \mathbb{E}_{\mathbf{x} \sim p_d(\mathbf{x})} \left[ -\partial_\psi E_\psi(\mathbf{x}) \right] + \mathbb{E}_{\mathbf{x} \sim h_{\psi,\theta}(\mathbf{x},\mathbf{z})} \left[ \partial_\psi E_\psi(\mathbf{x}) \right], \tag{7}$$

which decomposes into a positive and a negative phase, as discussed in Sec. 2.

**Reparametrized sampling in the negative phase:** For gradient estimation in the negative phase, we can draw samples from the model using MCMC. Naively, we can perform ancestral sampling, first sampling from the prior $p_\theta(\mathbf{z})$, then running MCMC for $p_\theta(\mathbf{x}|\mathbf{z})e^{-E_\psi(\mathbf{x})}$ in $\mathbf{x}$-space. This is problematic, since $p_\theta(\mathbf{x}|\mathbf{z})$ is often sharp and MCMC cannot mix when the conditioning $\mathbf{z}$ is fixed.

In this work, we instead run the MCMC iterations in the joint space of $\mathbf{z}$ and $\mathbf{x}$. Furthermore, we accelerate the sampling procedure using reparametrization for both $\mathbf{x}$ and the latent variables $\mathbf{z}$. Recall that when sampling from the VAE, we first sample $\mathbf{z} \sim p(\mathbf{z})$ and then $\mathbf{x} \sim p_\theta(\mathbf{x}|\mathbf{z})$. This sampling scheme can be reparametrized by sampling from a fixed noise distribution (e.g., $(\boldsymbol{\epsilon}_\mathbf{z}, \boldsymbol{\epsilon}_\mathbf{x}) \sim p_{\boldsymbol{\epsilon}} = \mathcal{N}(0, \mathbf{I})$) and deterministic transformations $T_\theta$ such that

$$\mathbf{z} = T_\theta^\mathbf{z}(\boldsymbol{\epsilon}_\mathbf{z}), \quad \mathbf{x} = T_\theta^\mathbf{x}(\mathbf{z}(\boldsymbol{\epsilon}_\mathbf{z}), \boldsymbol{\epsilon}_\mathbf{x}) = T_\theta^\mathbf{x}(T_\theta^\mathbf{z}(\boldsymbol{\epsilon}_\mathbf{z}), \boldsymbol{\epsilon}_\mathbf{x}). \tag{8}$$

Here, $T_\theta^\mathbf{z}$ denotes the transformation defined by the prior that transforms noise $\boldsymbol{\epsilon}_\mathbf{z}$ into prior samples $\mathbf{z}$ and $T_\theta^\mathbf{x}$ represents the decoder that transforms noise $\boldsymbol{\epsilon}_\mathbf{x}$ into samples $\mathbf{x}$, given prior samples $\mathbf{z}$. We can apply the same reparameterization when sampling from $h_{\psi,\theta}(\mathbf{x}, \mathbf{z})$. This corresponds to sampling $(\boldsymbol{\epsilon}_\mathbf{x}, \boldsymbol{\epsilon}_\mathbf{z})$ from the "base distribution":

$$h_{\psi,\theta}(\boldsymbol{\epsilon}_\mathbf{x}, \boldsymbol{\epsilon}_\mathbf{z}) \propto e^{-E_\psi(T_\theta^\mathbf{x}(T_\theta^\mathbf{z}(\boldsymbol{\epsilon}_\mathbf{z}), \boldsymbol{\epsilon}_\mathbf{x}))} p_{\boldsymbol{\epsilon}}(\boldsymbol{\epsilon}_\mathbf{x}, \boldsymbol{\epsilon}_\mathbf{z}), \tag{9}$$

and then transforming them to $\mathbf{x}$ and $\mathbf{z}$ via Eq. 8 (see Appendix B for derivation). Note that $\boldsymbol{\epsilon}_\mathbf{z}$ and $\boldsymbol{\epsilon}_\mathbf{x}$ have the same scale, as $p_{\boldsymbol{\epsilon}}(\boldsymbol{\epsilon}_\mathbf{x}, \boldsymbol{\epsilon}_\mathbf{z})$ is a standard Normal distribution, while the scales of $\mathbf{x}$ and $\mathbf{z}$ can be very different. Thus, running MCMC sampling with this reparameterization in the $(\boldsymbol{\epsilon}_\mathbf{x}, \boldsymbol{\epsilon}_\mathbf{z})$-space has the benefit that we do not need to tune the sampling scheme (e.g., step size in LD) for each variable. This is particularly helpful when $\mathbf{z}$ itself has multiple groups, as in our case.

**The advantages of two-stage training:** Besides avoiding the difficulties of estimating the full gradient of $\log Z_{\psi,\theta}$, two-stage training has additional advantages. As we discussed above, updating $\psi$ is computationally expensive, as each update requires an iterative MCMC procedure to draw samples from the model. The first stage of our training minimizes the distance between the VAE model and the data distribution, and in the second stage, the EBM further reduce the mismatch between the model and the data distribution. As the pre-trained VAE $p_\theta(\mathbf{x})$ provides a good approximation to $p_d(\mathbf{x})$ already, we expect that a relatively small number of expensive updates for training $\psi$ is needed. Moreover, the pre-trained VAE provides a latent space with an effectively lower dimensionality and a smoother distribution than the data distribution, which facilitates more efficient MCMC.

**Alternative extensions:** During the training of the energy function, we fix the VAE's parameters. In Appendix C, we discuss a possible extension to our training objective that also updates the VAE.

## 4 RELATED WORK

Early variants of EBMs include models whose energy is defined over both data and auxiliary latent variables (Salakhutdinov & Hinton, 2009; Hinton, 2012), and models using only data variables (Hinton, 2002; Mnih & Hinton, 2005). Their energy functions are simple and they do not scale to high

dimensional data. Recently, it was shown that EBMs with deep neural networks as energy function can successfully model complex data such as natural images (Du & Mordatch, 2019; Nijkamp et al., 2019b;a). They are trained with maximum likelihood and only model the data variable. Joint EBMs (Grathwohl et al., 2020a; Liu & Abbeel, 2020) model the joint distribution of data and labels. In contrast, our VAEBM models the joint distribution of data and general latent variables.

Besides fundamental maximum likelihood training, other techniques to train EBMs exist, such as minimizing F-divergence (Yu et al., 2020a) or Stein discrepancy (Grathwohl et al., 2020b), contrastive estimation (Gutmann & Hyvärinen, 2010; Gao et al., 2020) and denoising score matching (Li et al., 2019). Recently, noise contrastive score networks and diffusion models have demonstrated high quality image synthesis (Song & Ermon, 2019; 2020; Ho et al., 2020). These models are also based on denoising score matching (DSM) (Vincent, 2011), but do not parameterize any explicit energy function and instead directly model the vector-valued score function. We view score-based models as alternatives to EBMs trained with maximum likelihood. Although they do not require iterative MCMC during training, they need very long sampling chains to anneal the noise when sampling from the model ($\gtrsim 1000$ steps). Therefore, sample generation is extremely slow.

VAEBM is an EBM with a VAE component, and it shares similarities with work that builds connections between EBMs and other generative models. Zhao et al. (2017); Che et al. (2020); Song et al. (2020); Arbel et al. (2020) formulate EBMs with GANs, and use the discriminator to assign an energy. Xiao et al. (2020); Nijkamp et al. (2020) use normalizing flows that transport complex data to latent variables to facilitate MCMC sampling (Hoffman et al., 2019), and thus, their methods can be viewed as EBMs with flow component. However, due to their topology-preserving nature, normalizing flows cannot easily transport complex multimodal data, and their sample quality on images is limited. A few previous works combine VAEs and EBMs in different ways from ours. Pang et al. (2020) and Vahdat et al. (2018b;a; 2020) use EBMs for the prior distribution, and (Han et al., 2020; 2019) jointly learn a VAE and an EBM with independent sets of parameters by an adversarial game.

Finally, as we propose two-stage training, our work is related to post training of VAEs. Previous work in this direction learns the latent structure of pre-trained VAEs (Dai & Wipf, 2019; Xiao et al., 2019; Ghosh et al., 2020), and sampling from learned latent distributions improves sample quality. These methods cannot easily be extended to VAEs with hierarchical latent variables, as it is difficult to fit the joint distribution of multiple groups of variables. Our purpose for two-stage training is fundamentally different: we post-train an energy function to refine the distribution in data space.

## 5 EXPERIMENTS

In this section, we evaluate our proposed VAEBM through comprehensive experiments. Specifically, we benchmark sample quality in Sec. 5.1, provide detailed ablation studies on training techniques in Sec. 5.2, and study mode coverage of our model and test for spurious modes in Sec. 5.3. We choose NVAE (Vahdat & Kautz, 2020) as our VAE, which we pre-train, and use a simple ResNet as energy function $E_\psi$, similar to Du & Mordatch (2019). We draw approximate samples both for training and testing by running short Langevin dynamics chains on the distribution in Eq. 9. Note that in NVAE, the prior distribution is a group-wise auto-regressive Gaussian, and the conditional pixel-wise distributions in x are also Gaussian. Therefore, the reparameterization corresponds to shift and scale transformations. For implementation details, please refer to Appendix E.

### 5.1 IMAGE GENERATION

In Table 1, we quantitatively compare the sample quality of VAEBM with different generative models on (unconditional) CIFAR-10. We adopt Inception Score (IS) (Salimans et al., 2016) and FID (Heusel et al., 2017) as quantitative metrics. Note that FID reflects the sample quality more faithfully, as potential problems have been reported for IS on CIFAR-10 (Barratt & Sharma, 2018).

We observe that our VAEBM outperforms previous EBMs and other explicit likelihood-based models by a large margin. Note that introducing persistent chains during training only leads to slight improvement, while Du & Mordatch (2019) rely on persistent chains with a sample replay buffer. This is likely due to the efficiency of sampling in latent space. Our model also produces significantly better samples than NVAE, the VAE component of our VAEBM, implying a significant impact of our proposed energy-based refinement. We also compare our model with state-of-the-art GANs and

Table 1: IS and FID scores for unconditional generation on CIFAR-10.

|  | Model | IS↑ | FID↓ |
|---|---|---|---|
| **Ours** | VAEBM w/o persistent chain | 8.21 | 12.26 |
|  | VAEBM w/ persistent chain | 8.43 | 12.19 |
| **EBMs** | IGEBM (Du & Mordatch, 2019) | 6.02 | 40.58 |
|  | EBM with short-run MCMC (Nijkamp et al., 2019b) | 6.21 | - |
|  | F-div EBM (Yu et al., 2020a) | 8.61 | 30.86 |
|  | FlowCE (Gao et al., 2020) | - | 37.3 |
|  | FlowEBM (Nijkamp et al., 2020) | - | 78.12 |
|  | GEBM (Arbel et al., 2020) | - | 23.02 |
|  | Divergence Triangle (Han et al., 2020) | - | 30.1 |
| **Other Likelihood Models** | Glow (Kingma & Dhariwal, 2018) | 3.92 | 48.9 |
|  | PixelCNN (Oord et al., 2016b) | 4.60 | 65.93 |
|  | NVAE (Vahdat & Kautz, 2020) | 5.51 | 51.67 |
|  | VAE with EBM prior (Pang et al., 2020) | - | 70.15 |
| **Score-based Models** | NCSN (Song & Ermon, 2019) | 8.87 | 25.32 |
|  | NCSN v2 (Song & Ermon, 2020) | - | 31.75 |
|  | Multi-scale DSM (Li et al., 2019) | 8.31 | 31.7 |
|  | Denoising Diffusion (Ho et al., 2020) | 9.46 | 3.17 |
| **GAN-based Models** | SNGAN (Miyato et al., 2018) | 8.22 | 21.7 |
|  | SNGAN+DDLS (Che et al., 2020) | 9.09 | 15.42 |
|  | SNGAN+DCD (Song et al., 2020) | 9.11 | 16.24 |
|  | BigGAN (Brock et al., 2018) | 9.22 | 14.73 |
|  | StyleGAN2 w/o ADA (Karras et al., 2020a) | 8.99 | 9.9 |
| **Others** | PixelIQN (Ostrovski et al., 2018) | 5.29 | 49.46 |
|  | MoLM (Ravuri et al., 2018) | 7.90 | 18.9 |

recently proposed score-based models, and we obtain comparable or better results. Thus, we largely close the gap to GANs and score-models, while maintaining the desirable properties of models trained with maximum likelihood, such as fast sampling and better mode coverage.

Qualitative samples generated by our model are shown in Fig. 2a and intermediate samples along MCMC chains in Fig. 2b. We find that VAEBM generates good samples by running only a few MCMC steps. Initializing MCMC chains from the pre-trained VAE also helps quick equilibration.

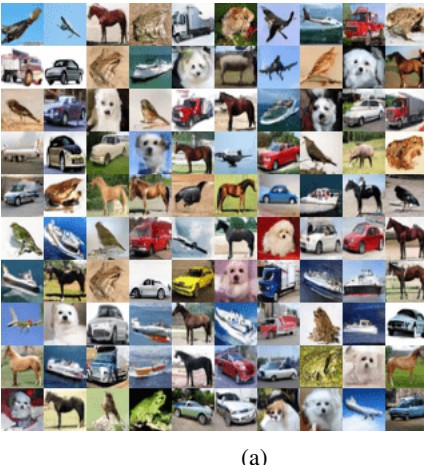
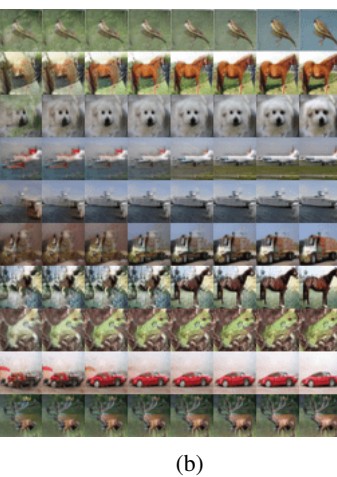

(a)  (b)

Figure 2: (a) CIFAR-10 samples generated by VAEBM. (b) Visualizing MCMC sampling chains. Samples are generated by running 16 LD steps. Chains are initialized with pre-trained VAE. We show intermediate samples at every 2 steps. See Appendix H for additional qualitative results.

We also train VAEBM on larger images, including CelebA 64, CelebA HQ 256 (Liu et al., 2015) and LSUN Church 64 (Yu et al., 2015). We report the FID scores for CelebA 64 and CelebA HQ 256 in Tables 2 and 3. On CelebA 64, our model obtains results comparable with the best GANs. Although our model obtains worse results than some advanced GANs on CelebA HQ 256, we significantly

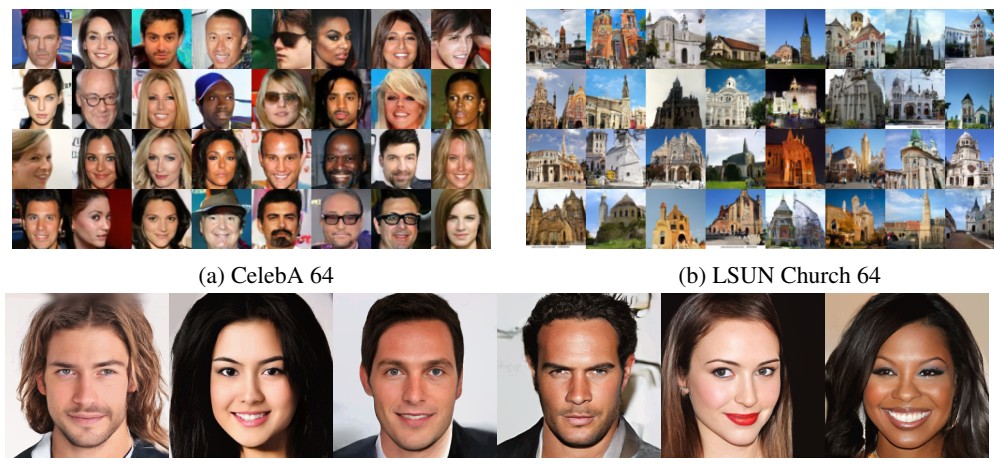

(a) CelebA 64             (b) LSUN Church 64

(c) CelebA HQ 256

Figure 3: Qualitative results on CelebA 64, LSUN Church 64 and CelebA HQ 256. For CelebA HQ 256, we initialize the MCMC chains with low temperature NVAE samples ($t = 0.7$) for better visual quality. On this dataset samples are selected for diversity. See Appendix H for additional qualitative results and uncurated CelebA HQ 256 samples obtained from higher temperature initializations. Note that the FID in Table 3 is computed with full temperature samples.

Table 2: Generative performance on CelebA 64

| Model | FID↓ |
|---|---|
| VAEBM (ours) | 5.31 |
| NVAE (Vahdat & Kautz) | 14.74 |
| Flow CE (Gao et al.) | 12.21 |
| Divergence Triangle (Han et al.) | 24.7 |
| NCSNv2 (Song & Ermon) | 26.86 |
| COCO-GAN (Lin et al.) | 4.0 |
| QA-GAN (Parimala & Channappayya) | 6.42 |

Table 3: Generative performance on CelebA HQ 256

| Model | FID↓ |
|---|---|
| VAEBM (ours) | 20.38 |
| NVAE (Vahdat & Kautz) | 45.11 |
| GLOW (Kingma & Dhariwal) | 68.93 |
| Advers. LAE (Pidhorskyi et al.) | 19.21 |
| PGGAN (Karras et al.) | 8.03 |

reduce the gap between likelihood based models and GANs on this dataset. On LSUN Church 64, we obtain FID 13.51, which significantly improves the NVAE baseline FID 41.3. We show qualitative samples in Fig. 3. Appendix H contains additional samples and MCMC visualizations.

Our model can produce impressive samples by running very short MCMC chains, however, we find that when we run longer MCMC chains than training chains, most chains stay around the local mode without traversing between modes. We believe that the non-mixing is due to the long mixing time of Langevin Dynamics Neal et al. (2011), as Nijkamp et al. (2019b;a) also observe that models trained with short-run MCMC have non-mixing long-run chains. We conjecture that mixing can be improved by training and sampling with more advanced MCMC techniques that are known to mix faster, such as HMC Neal et al. (2011), and this will be left for future work.

Table 4: Comparison for IS and FID on CIFAR-10 between several related training methods.

| Model | IS↑ | FID↓ |
|---|---|---|
| NVAE (Vahdat & Kautz) | 5.19 | 55.97 |
| EBM on $\mathbf{x}$ (Du & Mordatch) | 5.85 | 48.89 |
| EBM on $\mathbf{x}$, MCMC init w/ NVAE | 7.28 | 29.32 |
| WGAN w/ NVAE decoder | 7.41 | 20.39 |
| VAEBM (ours) | **8.15** | **12.96** |

Table 5: Mode coverage on StackedMNIST.

| Model | Modes↑ | KL↓ |
|---|---|---|
| VEEGAN (Srivastava et al.) | 761.8 | 2.173 |
| PacGAN (Lin et al.) | 992.0 | 0.277 |
| PresGAN (Dieng et al.) | 999.6 | 0.115 |
| InclusiveGAN (Yu et al.) | 997 | 0.200 |
| StyleGAN2 (Karras et al.) | 940 | 0.424 |
| VAEBM (ours) | **1000** | **0.087** |

## 5.2 ABLATION STUDIES

In Table 4, we compare VAEBM to several closely related baselines. All the experiments here are performed on CIFAR-10, and for simplicity, we use smaller models than those used in Table 1. Appendix F summarizes the experimental settings and Appendix G provides qualitative samples.

**Data space vs. augmented space:** One key difference between VAEBM and previous work such as Du & Mordatch (2019) is that our model is defined on the augmented space $(\mathbf{x}, \mathbf{z})$, while their EBM only involves $\mathbf{x}$. Since we pre-train the VAE, one natural question is whether our strong results are due to good initial samples $\mathbf{x}$ from the VAE, which are used to launch the MCMC chains. To address this, we train an EBM purely on $\mathbf{x}$ as done in Du & Mordatch (2019). We also train another EBM only on $\mathbf{x}$, but we initialize the MCMC chains with samples from the pre-trained NVAE instead of noise. As shown in line 3 of Table 4, this initialization helps the EBM which is defined only on $\mathbf{x}$. However, VAEBM in the augmented space outperforms the EBMs on $\mathbf{x}$ only by a large margin.

**Adversarial training vs. sampling:** The gradient for $\psi$ in Eq. 7 is similar to the gradient updates of WGAN's discriminator (Arjovsky et al., 2017). The key difference is that we draw (approximate) samples from $h_\psi(\mathbf{x})$ by MCMC, while WGAN draws negative samples from a generator (Che et al., 2020). WGAN updates the generator by playing an adversarial game, while we only update the energy function $E_\psi$. We compare these two methods by training $\psi$ and $\theta$ with the WGAN objective and initializing $\theta$ with the NVAE decoder. As shown in line 4 of Table 4, we significantly outperform the WGAN version of our model, implying the advantage of our method over adversarial training.

## 5.3 TEST FOR SPURIOUS OR MISSING MODES

We evaluate mode coverage on StackedMNIST. This dataset contains images generated by randomly choosing 3 MNIST images and stacking them along the RGB channels. Hence, the data distribution has 1000 modes. Following Lin et al. (2018), we report the number of covered modes and the KL divergence from the categorical distribution over 1000 categories from generated samples to true data (Table 5). VAEBM covers all modes and achieves the lowest KL divergence even compared to GANs that are specifically designed for this task. Hence, our model covers the modes more equally. We also plot the histogram of likelihoods for CIFAR-10 train/test images (Fig. 6, Appendix D) and present nearest neighbors of generated samples (Appendix I). We conclude that we do not overfit.

We evaluate spurious modes in our model by assessing its performance on out-of-distribution (OOD) detection. Specifically, we use VAEBM trained on CIFAR-10, and estimate unnormalized $\log h_{\psi,\theta}(\mathbf{x})$ on in-distribution samples (from CIFAR-10 test set) and OOD samples from several datasets. Following Nalisnick et al. (2019), we use area under the ROC curve (AUROC) as quantitative metric, where high AUROC indicates that the model correctly assigns low density to OOD samples. In Table 6, we see that VAEBM has significantly higher AUROC than NVAE, justifying our argument that the energy function reduces the likelihood of non-data-like regions. VAEBM also performs better than IGEBM and JEM, while worse than HDGE. However, we note that JEM and HDGE are classifier-based models, known to be better for OOD detection (Liang et al., 2018).

Table 6: Table for AUROC↑ of $\log p(\mathbf{x})$ computed on several OOD datasets. In-distribution dataset is CIFAR-10. Interp. corresponds to linear interpolation between CIFAR-10 images.

|  |  | SVHN | Interp. | CIFAR100 | CelebA |
|---|---|---|---|---|---|
| **Unsupervised Training** | NVAE (Vahdat & Kautz, 2020) | 0.42 | 0.64 | 0.56 | 0.68 |
|  | Glow (Kingma & Dhariwal, 2018) | 0.05 | 0.51 | 0.55 | 0.57 |
|  | IGEBM (Du & Mordatch, 2019) | 0.63 | **0.7** | 0.5 | 0.7 |
|  | Divergence Traingle (Han et al., 2020) | 0.68 | - | - | 0.56 |
|  | VAEBM (ours) | **0.83** | **0.7** | **0.62** | **0.77** |
| **Supervised Training** | JEM (Grathwohl et al., 2020a) | 0.67 | 0.65 | 0.67 | 0.75 |
|  | HDGE (Liu & Abbeel, 2020) | 0.96 | 0.82 | 0.91 | 0.8 |

## 5.4 EXACT LIKELIHOOD ESTIMATE ON 2D TOY DATA

VAEBM is an explicit likelihood model with a parameterized density function. However, like other energy-based models, the estimation of the exact likelihood is difficult due to the intractable partition

function $\log Z$. One possible way to estimate the partition function is to use Annealed Importance Sampling (AIS) (Neal, 2001). However, using AIS to estimate $\log Z$ in high-dimensional spaces is difficult. In fact, Du & Mordatch (2019) report that the estimation does not converge in 2 days on CIFAR-10. Furthermore, AIS gives a stochastic lower bound on $\log Z$, and therefore the likelihood computed with this estimated $\log Z$ would be an upper bound for the true likelihood. This makes the estimated likelihood hard to compare with the VAE's likelihood estimate, which is usually a lower bound on the true likelihood (Burda et al., 2015).

As a result, to illustrate that our model corrects the distribution learned by the VAE and improves the test likelihood, we conduct additional experiments on a 2-D toy dataset. We use the 25-Gaussians dataset, which is generated by a mixture of 25 two-dimensional isotropic Gaussian distributions arranged in a grid. This dataset is also studied in Che et al. (2020). The encoder and decoder of the VAE have 4 fully connected layers with 256 hidden units, and the dimension of the latent variables is 20. Our energy function has 4 fully connected layers with 256 hidden units.

In the 2-D domain, the partition function $\log Z$ can be accurately estimated by a numerical integration scheme. For the VAE, we use the IWAE bound (Burda et al., 2015) with 10,000 posterior samples to estimate its likelihood. We use 100,000 test samples from the true distribution to evaluate the likelihood. Our VAEBM obtains the average log likelihood of **-1.50** nats on test samples, which significantly improves the VAE, whose average test likelihood is **-2.97** nats. As a reference, we also analytically compute the log likelihood of test samples under the true distribution, and the result is **-1.10** nats.

We show samples from the true distribution, VAE and VAEBM in Figure 4. We observe that the VAEBM successfully corrects the distribution learned by the VAE and has better sample quality.

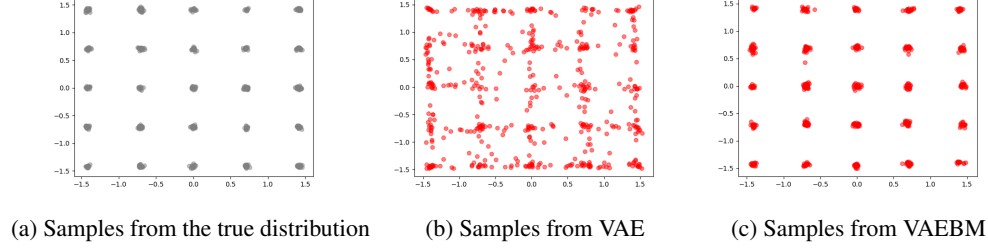

(a) Samples from the true distribution     (b) Samples from VAE     (c) Samples from VAEBM

Figure 4: Qualitative results on the 25-Gaussians dataset

## 5.5 SAMPLING EFFICIENCY

Despite their impressive sample quality, denoising score matching models (Song & Ermon, 2019; Ho et al., 2020) are slow at sampling, often requiring $\gtrsim 1000$ MCMC steps. Since VAEBM uses short MCMC chains, it takes only $8.79$ seconds to generate $50$ CIFAR-10 samples, whereas NCSN (Song & Ermon, 2019) takes $107.9$ seconds, which is about $12\times$ slower (see Appendix J for details).

## 6 CONCLUSIONS

We propose VAEBM, an energy-based generative model in which the data distribution is defined jointly by a VAE and an energy network, the EBM component of the model. In this joint model, the EBM and the VAE form a symbiotic relationship: the EBM component refines the initial VAE-defined distribution, while the VAE's latent embedding space is used to accelerate sampling from the joint model and therefore enables efficient training of the energy function. We show that our model can be trained effectively in two stages with a maximum likelihood objective and we can efficiently sample it by running short Langevin dynamics chains. Experimental results demonstrate strong generative performance on several image datasets. Future work includes further scaling up the model to larger images, applying it to other domains, and using more advanced sampling algorithms.

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

# A   DERIVING THE GRADIENT OF $\log Z_{\psi,\theta}$

Recall that $Z_{\psi,\theta} = \int p_\theta(\mathbf{x}) e^{-E_\psi(\mathbf{x})} d\mathbf{x}$. For the derivative of $\log Z_{\psi,\theta}$ w.r.t. $\theta$, we have:

$$
\begin{aligned}
\frac{\partial}{\partial \theta} \log Z_{\psi,\theta} &= \frac{\partial}{\partial \theta} \log \left( \int p_\theta(\mathbf{x}) e^{-E_\psi(\mathbf{x})} d\mathbf{x} \right) = \frac{1}{Z_{\psi,\theta}} \int \frac{\partial p_\theta(\mathbf{x})}{\partial \theta} e^{-E_\psi(\mathbf{x})} d\mathbf{x} \\
&= \frac{1}{Z_{\psi,\theta}} \int p_\theta(\mathbf{x}) e^{-E_\psi(\mathbf{x})} \frac{\partial \log p_\theta(\mathbf{x})}{\partial \theta} d\mathbf{x} = \int h_{\psi,\theta}(\mathbf{x}) \frac{\partial \log p_\theta(\mathbf{x})}{\partial \theta} d\mathbf{x} \\
&= \mathbb{E}_{\mathbf{x} \sim h_{\psi,\theta}(\mathbf{x},\mathbf{z})} \left[ \frac{\partial \log p_\theta(\mathbf{x})}{\partial \theta} \right]
\end{aligned}
\tag{10}
$$

Similarly, it is easy to show that $\frac{\partial}{\partial \psi} \log Z_{\psi,\theta} = \mathbb{E}_{\mathbf{x} \sim h_{\psi,\theta}(\mathbf{x},\mathbf{z})} \left[ -\frac{\partial E_\psi(\mathbf{x})}{\partial \psi} \right]$. Intuitively, both gradients encourage reducing the likelihood of the samples generated by the VAEBM model. Since, $h_{\psi,\theta}$ is an EBM, the expectation can be approximated using MCMC samples.

Note that Eq. 10 is further expanded to:

$$
\frac{\partial}{\partial \theta} \log Z_{\psi,\theta} = \mathbb{E}_{\mathbf{x} \sim h_{\psi,\theta}(\mathbf{x},\mathbf{z})} \left[ \mathbb{E}_{\mathbf{z}' \sim p_\theta(\mathbf{z}'|\mathbf{x})} \left[ \frac{\partial \log p_\theta(\mathbf{x},\mathbf{z}')}{\partial \theta} \right] \right],
$$

which can be approximated by first sampling from VAEBM using MCMC (i.e., $\mathbf{x} \sim h_{\psi,\theta}(\mathbf{x},\mathbf{z})$) and then sampling from the true posterior of the VAE (i.e., $\mathbf{z}' \sim p_\theta(\mathbf{z}'|\mathbf{x})$). The gradient term can be easily computed given the samples. Two approaches can be used to draw approximate samples from $p_\theta(\mathbf{z}'|\mathbf{x})$. i) We can replace $p_\theta(\mathbf{z}'|\mathbf{x})$ with the approximate posterior $q_\phi(\mathbf{z}'|\mathbf{x})$. However, the quality of this estimation depends on how well $q_\phi(\mathbf{z}'|\mathbf{x})$ matches the true posterior on samples generated by $h_{\psi,\theta}(\mathbf{x},\mathbf{z})$, which can be very different from the real data samples. To bring $q_\phi(\mathbf{z}'|\mathbf{x})$ closer to $p_\theta(\mathbf{z}'|\mathbf{x})$, we can maximize the variational bound (Eq. 3) on samples generated from $h_{\psi,\theta}(\mathbf{x},\mathbf{z})$ with respect to $\phi$, the encoder parameters[2]. However, this will add additional complexity to training. ii) Alternatively, we can use MCMC sampling to sample $\mathbf{z}' \sim p_\theta(\mathbf{z}'|\mathbf{x})$. To speed up MCMC, we can initialize the $\mathbf{z}'$ samples in MCMC with the original $\mathbf{z}$ samples that were drawn in the outer expectation (i.e., $\mathbf{x}, \mathbf{z} \sim h_{\psi,\theta}(\mathbf{x},\mathbf{z})$). However, with this approach, the computational complexity of the gradient estimation for the negative phase is doubled, as we now require running MCMC twice, once for $\mathbf{x}, \mathbf{z} \sim h_{\psi,\theta}(\mathbf{x},\mathbf{z})$ and again for $\mathbf{z}' \sim p_\theta(\mathbf{z}'|\mathbf{x})$.

We can entirely avoid the additional computational complexity and the complications of estimating $\frac{\partial}{\partial \theta} \log Z_{\psi,\theta}$, if we assume that the VAE is held fixed when training the EBM component of our VAEBM. This way, we require running MCMC only to sample $\mathbf{x} \sim h_{\psi,\theta}(\mathbf{x},\mathbf{z})$ to compute $\frac{\partial}{\partial \psi} \log Z_{\psi,\theta}$.

# B   REPARAMETRIZATION FOR EBM

Suppose we draw the re-parametrization variables $(\boldsymbol{\epsilon}_{\mathbf{x}}, \boldsymbol{\epsilon}_{\mathbf{z}}) \sim p_{\boldsymbol{\epsilon}}(\boldsymbol{\epsilon}_{\mathbf{x}}, \boldsymbol{\epsilon}_{\mathbf{z}})$. For convenience, we denote

$$
T_\theta(\boldsymbol{\epsilon}_{\mathbf{x}}, \boldsymbol{\epsilon}_{\mathbf{z}}) = (T_\theta^{\mathbf{x}}(T_\theta^{\mathbf{z}}(\boldsymbol{\epsilon}_{\mathbf{z}}), \boldsymbol{\epsilon}_{\mathbf{x}}), T_\theta^{\mathbf{z}}(\boldsymbol{\epsilon}_{\mathbf{z}})) = (\mathbf{x}, \mathbf{z}).
\tag{11}
$$

Since $T_\theta$ is a deterministic and invertible transformation that maps $(\boldsymbol{\epsilon}_{\mathbf{x}}, \boldsymbol{\epsilon}_{\mathbf{z}})$ to $(\mathbf{x}, \mathbf{z})$, by the change of variables formula, we can write

$$
p_\theta(\mathbf{x}, \mathbf{z}) = p_{\boldsymbol{\epsilon}}(T_\theta^{-1}(\mathbf{x}, \mathbf{z})) \left| \det \left( J_{T_\theta^{-1}}(\mathbf{x}, \mathbf{z}) \right) \right|,
\tag{12}
$$

where $J_{T_\theta^{-1}}$ is the Jacobian of $T_\theta^{-1}$. Consider a Gaussian distribution as a simple example: if $\mathbf{z} \sim \mathcal{N}(\mu_{\mathbf{z}}, \sigma_{\mathbf{z}})$ and $\mathbf{x}|\mathbf{z} \sim \mathcal{N}(\mu_{\mathbf{x}}(\mathbf{z}), \sigma_{\mathbf{x}}(\mathbf{z}))$, then

$$
\mathbf{z} = T_\theta^{\mathbf{z}}(\boldsymbol{\epsilon}_{\mathbf{z}}) = \mu_{\mathbf{z}} + \sigma_{\mathbf{z}} \cdot \boldsymbol{\epsilon}_{\mathbf{z}}, \quad \mathbf{x} = T_\theta^{\mathbf{x}}(\boldsymbol{\epsilon}_{\mathbf{x}}, \boldsymbol{\epsilon}_{\mathbf{z}}) = \mu_{\mathbf{x}}(\mathbf{z}) + \sigma_{\mathbf{x}}(\mathbf{z}) \cdot \boldsymbol{\epsilon}_{\mathbf{x}},
$$

and

$$
J_{T_\theta^{-1}}(\mathbf{x}, \mathbf{z}) = [\sigma_{\mathbf{x}}(\mathbf{z})^{-1}, \sigma_{\mathbf{z}}^{-1}].
$$

---

[2]Maximizing ELBO with respect to $\phi$ corresponds to minimizing $D_{\mathrm{KL}}(q_\phi(\mathbf{z}|\mathbf{x})||p_\theta(\mathbf{z}|\mathbf{x}))$ while $\theta$ is fixed.

Recall that the generative model of our EBM is

$$h_{\psi,\theta}(\mathbf{x}, \mathbf{z}) = \frac{e^{-E_\psi(\mathbf{x})} p_\theta(\mathbf{x}, \mathbf{z})}{Z_{\psi,\theta}}. \tag{13}$$

We can apply the change of variable to $h_{\psi,\theta}(\mathbf{x}, \mathbf{z})$ in similar manner:

$$h_{\psi,\theta}(\boldsymbol{\epsilon}_\mathbf{x}, \boldsymbol{\epsilon}_\mathbf{z}) = h_{\psi,\theta}(T_\theta(\boldsymbol{\epsilon}_\mathbf{x}, \boldsymbol{\epsilon}_\mathbf{z})) \left| \det\left(J_{T_\theta}\left(\boldsymbol{\epsilon}_\mathbf{x}, \boldsymbol{\epsilon}_\mathbf{z}\right)\right)\right|, \tag{14}$$

where $J_{T_\theta}$ is the Jacobian of $T_\theta$.

Since we have the relation

$$J_{\mathbf{f}^{-1}} \circ \mathbf{f} = J_\mathbf{f}^{-1} \tag{15}$$

for invertible function $\mathbf{f}$, we have that

$$h_{\psi,\theta}(\boldsymbol{\epsilon}_\mathbf{x}, \boldsymbol{\epsilon}_\mathbf{z}) = h_{\psi,\theta}(T_\theta(\boldsymbol{\epsilon}_\mathbf{x}, \boldsymbol{\epsilon}_\mathbf{z})) \left| \det\left(J_{T_\theta}\left(\boldsymbol{\epsilon}_\mathbf{z}, \boldsymbol{\epsilon}_\mathbf{x}\right)\right)\right| \tag{16}$$

$$= \frac{1}{Z_{\psi,\theta}} e^{-E_\psi(T_\theta(\boldsymbol{\epsilon}_\mathbf{x}, \boldsymbol{\epsilon}_\mathbf{z}))} p_\theta(T_\theta(\boldsymbol{\epsilon}_\mathbf{x}, \boldsymbol{\epsilon}_\mathbf{z})) \left| \det\left(J_{T_\theta}\left(\boldsymbol{\epsilon}_\mathbf{x}, \boldsymbol{\epsilon}_\mathbf{z}\right)\right)\right| \tag{17}$$

$$= \frac{1}{Z_{\psi,\theta}} e^{-E_\psi(T_\theta(\boldsymbol{\epsilon}_\mathbf{x}, \boldsymbol{\epsilon}_\mathbf{z}))} p_{\boldsymbol{\epsilon}}(T_\theta^{-1}(\mathbf{x}, \mathbf{z})) \left| \det\left(J_{T_\theta^{-1}}\left(\mathbf{x}, \mathbf{z}\right)\right)\right| \left| \det\left(J_{T_\theta}\left(\boldsymbol{\epsilon}_\mathbf{x}, \boldsymbol{\epsilon}_\mathbf{z}\right)\right)\right| \tag{18}$$

$$= \frac{1}{Z_{\psi,\theta}} e^{-E_\psi(T_\theta(\boldsymbol{\epsilon}_\mathbf{x}, \boldsymbol{\epsilon}_\mathbf{z}))} p_{\boldsymbol{\epsilon}}(T_\theta^{-1}(\mathbf{x}, \mathbf{z})) \tag{19}$$

$$= \frac{1}{Z_{\psi,\theta}} e^{-E_\psi(T_\theta(\boldsymbol{\epsilon}_\mathbf{x}, \boldsymbol{\epsilon}_\mathbf{z}))} p_{\boldsymbol{\epsilon}}(\boldsymbol{\epsilon}_\mathbf{x}, \boldsymbol{\epsilon}_\mathbf{z}), \tag{20}$$

which is the distribution in Eq. 9.

After we obtained samples $(\boldsymbol{\epsilon}_\mathbf{x}, \boldsymbol{\epsilon}_\mathbf{z})$ from the distribution in Eq. 20, we obtain $(\mathbf{x}, \mathbf{z})$ by applying the transformation $T_\theta$ in Eq. 11.

## B.1 COMPARISON OF SAMPLING IN $(\boldsymbol{\epsilon}_\mathbf{x}, \boldsymbol{\epsilon}_\mathbf{z})$-SPACE AND IN $(\mathbf{x}, \mathbf{z})$-SPACE

Above we show that sampling from $h_{\psi,\theta}(\mathbf{x}, \mathbf{z})$ is equivalent to sampling from $h_{\psi,\theta}(\boldsymbol{\epsilon}_\mathbf{x}, \boldsymbol{\epsilon}_\mathbf{z})$ and applying the appropriate variable transformation. Here, we further analyze the connections between sampling from these two distributions with Langevin dynamics. Since each component of $\mathbf{x}$ and $\mathbf{z}$ can be re-parametrzied with scaling and translation of standard Gaussian noise, without loss of generality, we assume a variable $\mathbf{c}$ ($\mathbf{c}$ can be a single latent variable in $\mathbf{z}$ or a single pixel in $\mathbf{x}$) and write

$$\mathbf{c} = \mu + \sigma\boldsymbol{\epsilon}.$$

Suppose we sample in the $\boldsymbol{\epsilon}$ space with energy function $f$ on $\mathbf{c}$ and step size $\eta$. The update for $\boldsymbol{\epsilon}$ is

$$\boldsymbol{\epsilon}_{t+1} = \boldsymbol{\epsilon}_t - \frac{\eta}{2}\nabla_{\boldsymbol{\epsilon}}f + \sqrt{\eta}\omega_t, \quad \omega_t \sim \mathcal{N}(0, \mathbf{I}).$$

Now we plug $\boldsymbol{\epsilon}_{t+1}$ into the expression of $\mathbf{c}$ while noting that $\nabla_{\boldsymbol{\epsilon}}f = \sigma\nabla_{\mathbf{c}}f$. We obtain

$$\mathbf{c}_{t+1} = \mu + \sigma\boldsymbol{\epsilon}_{t+1} = \mu + \sigma\left(\boldsymbol{\epsilon}_t - \frac{\eta}{2}\nabla_{\boldsymbol{\epsilon}}f + \sqrt{\eta}\omega_t\right)$$

$$= \mu + \sigma\boldsymbol{\epsilon}_t - \frac{\sigma^2\eta}{2}\nabla_{\mathbf{c}}f + \sqrt{\eta\sigma^2}\omega_t$$

$$= \mathbf{c}_t - \frac{\sigma^2\eta}{2}\nabla_{\mathbf{c}}f + \sqrt{\eta\sigma^2}\omega_t.$$

Therefore, we see that running Langevin dynamics in $(\boldsymbol{\epsilon}_\mathbf{x}, \boldsymbol{\epsilon}_\mathbf{z})$-space is equivalent to running Langevin dynamics in $(\mathbf{x}, \mathbf{z})$-space with step size for each component of $\mathbf{z}$ and $\mathbf{x}$ adjusted by its variance. However, considering the high dimensionality of $\mathbf{x}$ and $\mathbf{z}$, the step size adjustment is difficult to implement.

The analysis above only considers a variable individually. More importantly, our latent variable $\mathbf{z}$ in the prior follows block-wise auto-regressive Gaussian distributions, so the variance of each

component in $\mathbf{z}_i$ depends on the value of $\mathbf{z}_{<i}$. We foresee that because of this dependency, using a fixed step size per component of $\mathbf{z}$ will not be effective, even when it is set differently for each component. In contrast, all the components in $(\boldsymbol{\epsilon}_{\mathbf{x}}, \boldsymbol{\epsilon}_{\mathbf{z}})$-space have a unit variance. Hence, a universal step size for all the variables in this space can be used.

To further provide empirical evidence that adjusting the step size for each variable is necessary, we try sampling directly in $(\mathbf{x}, \mathbf{z})$-space without adjusting the step size (i.e., use a universal step size for all variables). Qualitative results are presented in Figure 5. We examine several choices for the step size and we cannot obtain high-quality samples.

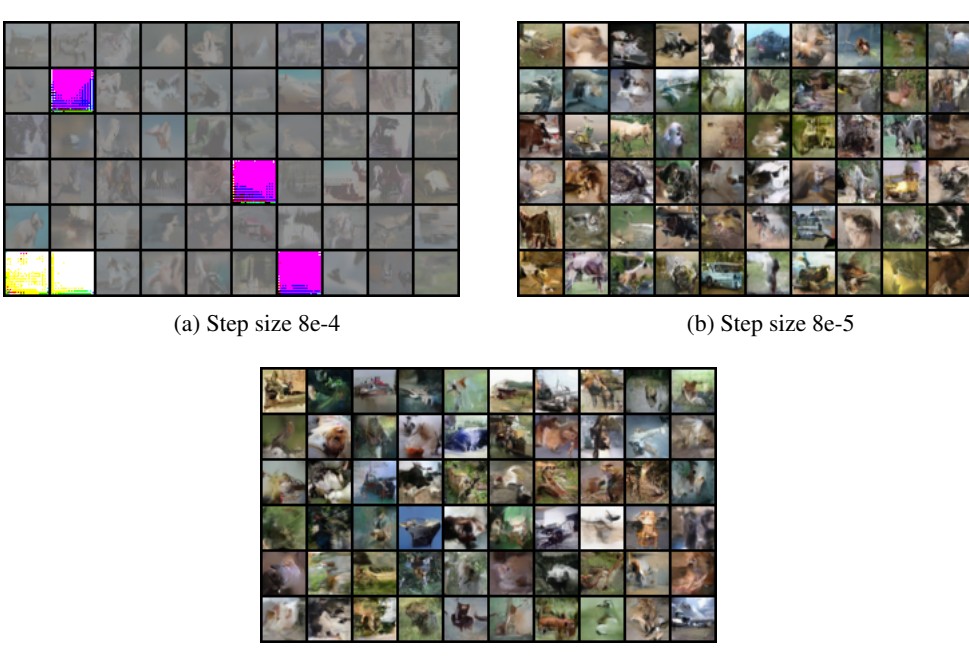

(a) Step size 8e-4          (b) Step size 8e-5

(c) Step size 8e-6

Figure 5: Qualitative samples obtained from sampling in $(\mathbf{x}, \mathbf{z})$-space with different step sizes.

In conclusion, the re-parameterization provides an easy implementation to adjust step size for each variable, and the adjustment is shown to be crucial to obtain good samples.

## C  EXTENSION TO TRAINING OBJECTIVE

In the first stage of training VAEBM, the VAE model is trained by maximizing the training data log-likelihood which corresponds to minimizing an upper bound on $D_{\mathrm{KL}}(p_d(\mathbf{x})||p_\theta(\mathbf{x}))$ w.r.t. $\theta$. In the second stage, when we are training the EBM component, we use the VAE model to sample from the joint VAEBM by running the MCMC updates in the joint space of $\boldsymbol{\epsilon}_{\mathbf{z}}$ and $\boldsymbol{\epsilon}_{\mathbf{x}}$. Ideally, we may want to bring $p_\theta(\mathbf{x})$ closer to $h_{\psi,\theta}(\mathbf{x})$ in the second stage, because when $p_\theta(\mathbf{x}) = h_{\psi,\theta}(\mathbf{x})$, we will not need the expensive updates for $\psi$. We can bring $p_\theta(\mathbf{x})$ closer to $h_{\psi,\theta}(\mathbf{x})$ by minimizing $D_{\mathrm{KL}}(p_\theta(\mathbf{x})||h_{\psi,\theta}(\mathbf{x}))$ with respect to $\theta$ which was recently discussed in the context of an EBM-interpretation of GANs by Che et al. (2020). To do so, we assume the target distribution $h_{\psi,\theta}(\mathbf{x})$ is fixed and create a copy of $\theta$, named $\theta'$, and we update $\theta'$ by the gradient:

$$\nabla_{\theta'} D_{\mathrm{KL}}(p_{\theta'}(\mathbf{x})||h_{\psi,\theta}(\mathbf{x})) = \nabla_{\theta'} \mathbb{E}_{\mathbf{x} \sim p_{\theta'}(\mathbf{x})} \left[ E_\psi(\mathbf{x}) \right] \tag{21}$$

In other words, one update step for $\theta'$ that minimizes $D_{\mathrm{KL}}(p'_\theta(\mathbf{x})||h_{\psi,\theta}(\mathbf{x}))$ w.r.t. $\theta'$ can be easily done by drawing samples from $p'_\theta(\mathbf{x})$ and minimizing the energy-function w.r.t. $\theta'$. Note that this approach is similar to the generator update in training Wasserstein GANs (Arjovsky et al., 2017). The above KL objective will encourage $p_\theta(\mathbf{x})$ to model dominants modes in $h_{\psi,\theta}(\mathbf{x})$. However, it may cause $p_\theta(\mathbf{x})$ to drop modes.

## C.1 DERIVATION

Our derivation largely follows Appendix A.2 of Che et al. (2020). Note that every time we update $\theta$, we are actually taking the gradient w.r.t $\theta'$, which can be viewed as a copy of $\theta$ and is initialized as $\theta$. In particular, we should note that the $\theta$ in $h_{\psi,\theta}(\mathbf{x})$ is fixed. Therefore, we have

$$
\nabla_{\theta'} D_{\mathrm{KL}}(p_{\theta'}(\mathbf{x})||h_{\psi,\theta}(\mathbf{x})) = \nabla_{\theta'} \int p_{\theta'}(\mathbf{x}) \left[\log p_{\theta'}(\mathbf{x}) - \log h_{\psi,\theta}(\mathbf{x})\right] d\mathbf{x}
$$

$$
= \int \left[\nabla_{\theta'} p_{\theta'}(\mathbf{x})\right] \left[\log p_{\theta'}(\mathbf{x}) - \log h_{\psi,\theta}(\mathbf{x})\right] d\mathbf{x}
$$

$$
+ \underbrace{\int p_{\theta'}(\mathbf{x}) \left[\nabla_{\theta'} \log p_{\theta'}(\mathbf{x}) - \nabla_{\theta'} \log h_{\psi,\theta}(\mathbf{x})\right] d\mathbf{x}}_{=0} \quad (22)
$$

$$
= \int \left[\nabla_{\theta'} p_{\theta'}(\mathbf{x})\right] \left[\log p_{\theta'}(\mathbf{x}) - \log h_{\psi,\theta}(\mathbf{x})\right] d\mathbf{x}, \quad (23)
$$

where the second term in Eq. 22 is 0 because the $\log h_{\psi,\theta}(\mathbf{x})$ does not depend on $\theta'$ and the expectation of the score function is 0:

$$
\int p_{\theta'}(\mathbf{x}) \nabla_{\theta'} \log p_{\theta'}(\mathbf{x}) d\mathbf{x} = \mathbb{E}_{\mathbf{x} \sim p_{\theta'}(\mathbf{x})} \left[\nabla_{\theta'} \log p_{\theta'}(\mathbf{x})\right] = 0.
$$

Recall that $\theta'$ has the same value as $\theta$ before the update, so

$$
\log p_{\theta'}(\mathbf{x}) - \log h_{\psi,\theta}(\mathbf{x}) = \log \left[\frac{p_{\theta'}(\mathbf{x})}{p_\theta(\mathbf{x}) e^{-E_\psi(\mathbf{x})}}\right] + \log Z_{\psi,\theta}
$$

$$
= E_\psi(\mathbf{x}) + \log Z_{\psi,\theta}. \quad (24)
$$

Plug Eq. 24 into Eq. 23, we have

$$
\nabla_{\theta'} D_{\mathrm{KL}}(p_{\theta'}(\mathbf{x})||h_{\psi,\theta}(\mathbf{x})) = \int \nabla_{\theta'} p_{\theta'}(\mathbf{x}) \left[E_\psi(\mathbf{x}) + \log Z_{\psi,\theta}\right] d\mathbf{x}
$$

$$
= \nabla_{\theta'} \mathbb{E}_{\mathbf{x} \sim p_{\theta'}(\mathbf{x})} \left[E_\psi(\mathbf{x})\right], \quad (25)
$$

since

$$
\int \nabla_{\theta'} p_{\theta'}(\mathbf{x}) \log Z_{\psi,\theta} d\mathbf{x} = \nabla_{\theta'} \log Z_{\psi,\theta} \int p_{\theta'}(\mathbf{x}) d\mathbf{x} = \nabla_{\theta'} \log Z_{\psi,\theta} = 0.
$$

## C.2 RESULTS

We train VAEBM with an additional loss term that updates the parameter $\theta$ to minimize $D_{\mathrm{KL}}(p_\theta(\mathbf{x})||h_{\psi,\theta}(\mathbf{x}))$ as explained above. Our experiment uses the same initial VAE as in Sec. 5.2, and details of the implementation are introduced in Appendix F. We obtain FID 14.0 and IS 8.05, which is similar to the results of plain VAEBM (FID 12.96 and IS 8.15). Therefore, we conclude that training the model by minimizing $D_{\mathrm{KL}}(p_\theta(\mathbf{x})||h_{\psi,\theta}(\mathbf{x}))$ does not improve the performance, and updating the decoder is not necessary. This is likely because the initial VAE is pulled as closely as possible to the data distribution already, which is also the target for the joint VAEBM $h_{\psi,\theta}(\mathbf{x})$.

## D COMPARING LIKELIHOODS ON TRAIN AND TEST SET

In Figure 6, we plot a histogram of unnormalized log-likelihoods of 10k CIFAR-10 train set and test set images. We see that our model assigns similar likelihoods to both train and test set images. This indicates that VAEBM generalizes well to unseen data and covers modes in the training data well.

## E IMPLEMENTATION DETAILS

In this section, we introduce the details of training and sampling from VAEBM.

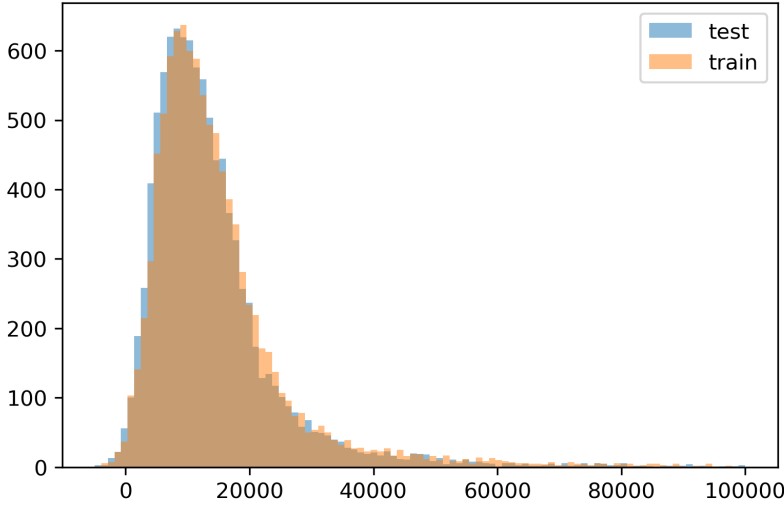

Figure 6: Histogram of unnormalized log-likelihoods on 10k CIFAR-10 train and test set images

**NVAE:** VAEBM uses NVAE as the $p_\theta(\mathbf{x})$ component in the model. We train the NVAE with its official implementation[3]. We largely follow the default settings, with one major difference that we use a Gaussian decoder instead of a discrete logistic mixture decoder as in Vahdat & Kautz (2020). The reason for this is that we can run Langevin dynamics only with continuous variables. The number of latent variable groups for CIFAR-10, CelebA 64, LSUN Church 64 and CelebA HQ 256 are 30, 15, 15 and 20, respectively.

Table 7: Network structures for the energy function $E_\psi(\mathbf{x})$

| CIFAR-10 | CelebA 64 | LSUN Church 64 | CelebA HQ 256 |
|---|---|---|---|
| $3 \times 3$ conv2d, 128 | $3 \times 3$ conv2d, 64 | $3 \times 3$ conv2d, 64 | $3 \times 3$ conv2d, 64 |
| ResBlock down 128 | ResBlock down 64 | ResBlock down 64 | ResBlock down 64 |
| ResBlock 128 | ResBlock 64 | ResBlock 64 | ResBlock 64 |
| ResBlock down 256 | ResBlock down 128 | ResBlock down 128 | ResBlock down 128 |
| ResBlock 256 | ResBlock 128 | ResBlock 128 | ResBlock 128 |
| ResBlock down 256 | ResBlock down 128 | ResBlock 128 | ResBlock down 128 |
| ResBlock 256 | ResBlock down 256 | ResBlock down 128 | ResBlock 128 |
| Global Sum Pooling | ResBlock 256 | ResBlock 256 | ResBlock down 256 |
| FC layer → scalar | Global Sum Pooling | ResBlock down 256 | ResBlock 256 |
|  | FC layer → scalar | ResBlock 256 | ResBlock down 256 |
|  |  | Global Sum Pooling | ResBlock 256 |
|  |  | FC layer → scalar | ResBlock down 512 |
|  |  |  | ResBlock 512 |
|  |  |  | Global Sum Pooling |
|  |  |  | FC layer → scalar |

**Network for energy function:** We largely adopt the energy network structure for CIFAR-10 in Du & Mordatch (2019), and we increase the depth of the network for larger images. There are 2 major differences between our energy networks and the ones used in Du & Mordatch (2019): **1.** we replace the LeakyReLU activations with Swish activations, as we found it improves training stability, and **2.** we do not use spectral normalization (Miyato et al., 2018); instead, we use weight normalization with data-dependent initialization (Salimans & Kingma, 2016). The network structure for each dataset is presented in Table 7.

**Training of energy function:** We train the energy function by minimizing the negative log likelihood and an additional spectral regularization loss which penalizes the spectral norm of each convolutional layer in $E_\psi$. The spectral regularization loss is also used in training NVAE, as we found

---

[3]https://github.com/NVlabs/NVAE

it helpful to regularize the sharpness of the energy network and better stabilize training. We use a coefficient 0.2 for the spectral regularization loss.

Table 8: Important hyper-parameters for training VAEBM

| Dataset | Learning rate | Batch size | Persistent | # of LD steps | LD Step size |
|---|---|---|---|---|---|
| CIFAR-10 w/o persistent chain | $4e-5$ | 32 | No | 10 | $8e-5$ |
| CIFAR-10 w/ persistent chain | $4e-5$ | 32 | Yes | 6 | $6e-5$ |
| CelebA 64 | $5e-5$ | 32 | No | 10 | $5e-6$ |
| LSUN Church 64 | $4e-5$ | 32 | Yes | 10 | $4e-6$ |
| CelebA HQ 256 | $4e-5$ | 16 | Yes | 6 | $3e-6$ |

We summarize some key hyper-parameters we used to train VAEBM in Table 8.

On all datasets, we train VAEBM using the Adam optimizer (Kingma & Ba, 2015) and weight decay $3e-5$. We use constant learning rates, shown in Table 8. Following Du & Mordatch (2019), we clip training gradients that are more than 3 standard deviations from the 2nd-order Adam parameters.

While persistent sampling using a sample replay buffer has little effect on CIFAR-10, we found it to be useful on large images such as CelebA HQ 256. When we do not use persistent sampling, we always initialize the LD chains with $(\epsilon_{\mathbf{x}}, \epsilon_{\mathbf{z}})$, sampled from a standard Gaussian. When we use persistent sampling in training, we keep a sample replay buffer that only stores samples of $\epsilon_{\mathbf{z}}$, while $\epsilon_{\mathbf{x}}$ is always initialized from a standard Gaussian. The size of the replay buffer is 10,000 for CIFAR-10 and LSUN Church 64, and 8,000 for CelebA HQ 256. At every training iteration, we initialize the MCMC chains on $\epsilon_{\mathbf{z}}$ by drawing $\epsilon_{\mathbf{z}}$ from the replay buffer with probability $p$ and from standard Gaussian with probability $1 - p$. For CIFAR-10 and LSUN Church 64, we linearly increase $p$ from 0 to 0.6 in 5,000 training iterations, and for CelebA HQ 256, we linearly increase $p$ from 0 to 0.6 in 3,000 training iterations. The settings of Langevin dynamics are presented in Table 8.

We do not explicitly set the number of training iterations. Instead, we follow Du & Mordatch (2019) to train the energy network until we cannot generate realistic samples anymore. This happens when the model overfits the training data and hence energies of negative samples are much larger than energies of training data. Typically, training takes around 25,000 iterations (or 16 epochs) on CIFAR-10, 20,000 iterations (or 3 epochs) on CelebA 64, 20,000 iterations (or 5 epochs) on LSUN Church 64, and 9,000 iterations (or 5 epochs) on CelebA HQ 256.

**Test time sampling:** After training the model, we generate samples for evaluation by running Langvin dynamics with $(\epsilon_{\mathbf{x}}, \epsilon_{\mathbf{z}})$ initialized from standard Gaussian, regardless of whether persistent sampling is used in training or not. We run slightly longer LD chains than training to obtain the best sample quality. In particular, our reported values are obtained from running 16 steps of LD for CIFAR-10, 20 steps of LD for CelebA64 and LSUN Church 64, and 24 steps for CelebA HQ 256. The step sizes are the same as training step sizes.

In CelebA HQ 256 dataset, we optionally use low temperature initialization for better visual quality. To do this, we first draw samples from the VAE with low temperature and readjusted the BN statistics as introduced by Vahdat & Kautz (2020), and then initialize the MCMC chain by $(\epsilon_{\mathbf{x}}, \epsilon_{\mathbf{z}})$ obtained by encoding the low-temperature samples using VAE's encoder without readjusted BN statistics.

**Evaluation metrics:** We use the official implementations of FID[4] and IS[5]. We compute IS using 50k CIFAR 10 samples, and we compute FID between 50k generated samples and training images, except for CelebA HQ 256 where we use 30k training images (the CelebA HQ dataset contains only 30k samples).

## F    SETTINGS FOR ABLATION STUDY

In this section, we present the details of ablation experiments in Sec. 5.2. Throughout ablation experiments, we use a smaller NVAE with 20 groups of latent variables trained on CIFAR-10. We use the same network architectures for the energy network as in Table 7, with potentially different

---

[4]https://github.com/bioinf-jku/TTUR
[5]https://github.com/openai/improved-gan/tree/master/inception_score

normalization techniques discussed below. We spent significant efforts on improving each method we compare against, and we report the settings that led to the best results.

**WGAN initialized with NVAE decoder:** We initialize the generator with the pre-trained NVAE decoder, and the discriminator is initialized by a CIFAR-10 energy network with random weights. We use spectral normalization and batch normalization in the discriminator as we found them necessary for convergence. We update the discriminator using the Adam optimizer with constant learning rate $5\mathrm{e}{-5}$, and update the generator using the Adam optimizer with initial learning rate $5\mathrm{e}{-6}$ and cosine decay schedule. We train the generator and discriminator for 40k iterations, and we reach convergence of sample quality towards the end of training.

**EBM on x, w/ or w/o initializing MCMC with NVAE samples:** We train two EBMs on data space similar to Du & Mordatch (2019), where for one of them, we use the pre-trained NVAE to initialize the MCMC chains that draw samples during training. The setting for training these two EBMs are the same except for the initialization of MCMC. We use spectral normalization in the energy network and energy regularization in the training objective as done in Du & Mordatch (2019) because we found these modifications to improve performance. We train the energy function using the Adam optimizer with constant learning rate $1\mathrm{e}{-4}$. We train for 100k iterations, and we reach convergence of sample quality towards the end of training. During training, we draw samples from the model following the MCMC settings in Du & Mordatch (2019). In particular, we use persistent sampling and sample from the sample replay buffer with probability 0.95. We run 60 steps of Langevin dynamics to generate negative samples and we clip gradients to have individual value magnitudes of less than 0.01. We use a step size of 10 for each step of Langevin dynamics. For test time sampling, we generate samples by running 150 steps of LD with the same settings as during training.

**VAEBM with $D_{\mathrm{KL}}(p_\theta(\mathbf{x})||h_{\psi,\theta}(\mathbf{x}))$ loss:** We use the same network structure for $E_\psi$ as in VAEBM. We find persistent sampling significantly hurts the performance in this case, possibly due to the fact that the decoder is updated and hence the initial samples from the decoder change throughout training. Therefore, we do not use persistent training. We train the energy function using the Adam optimizer with constant learning rate $5\mathrm{e}{-5}$. We draw negative samples by running 10 steps of LD with step size $8\mathrm{e}{-5}$. We update the decoder with the gradient in Eq. 21 using the Adam optimizer with initial learning rate $5\mathrm{e}{-6}$ and cosine decay schedule. For test time sampling, we run 15 steps of LD with step size $5\mathrm{e}{-6}$.

**VAEBM:** The training of VAEBM in this section largely follows the settings described in Appendix E. We use the same energy network as for CIFAR-10, and we train using the Adam optimizer with constant learning rate $5\mathrm{e}{-5}$. Again, we found that the performance of VAEBM with or without persistent sampling is similar. We adopt persistent sampling in this section because it is faster. The setting for the buffer is the same as in Appendix E. We run 5 steps of LD with step size $8\mathrm{e}{-5}$ during training, and 15 steps of LD with the same step size in testing.

## G    QUALITATIVE RESULTS OF ABLATION STUDY

In Figure 7, we show qualitative samples from models corresponding to each item in Table 4, as well as samples generated by VAEBM with additional $D_{\mathrm{KL}}(p_\theta(\mathbf{x})||h_{\psi,\theta}(\mathbf{x}))$ loss.

## H    ADDITIONAL QUALITATIVE RESULTS

We present additional qualitative results in this section.

Additional samples and visualizations of MCMC on CIFAR-10 are in Figures 8 and 9, respectively.

Additional samples on CelebA 64 are in Figure 10.

Additional samples on LSUN Church 64 are in Figure 11. We visualize the effect of running MCMC by displaying sample pairs before and after MCMC in Figure 12.

Additional samples on CelebA HQ 256 generated by initializing VAE samples with temperature 0.7 are shown in Figure 13. Samples generated by initializing VAE samples with full temperature 1.0 are shown in Figure 14. We visualize the effect of running MCMC by displaying sample pairs

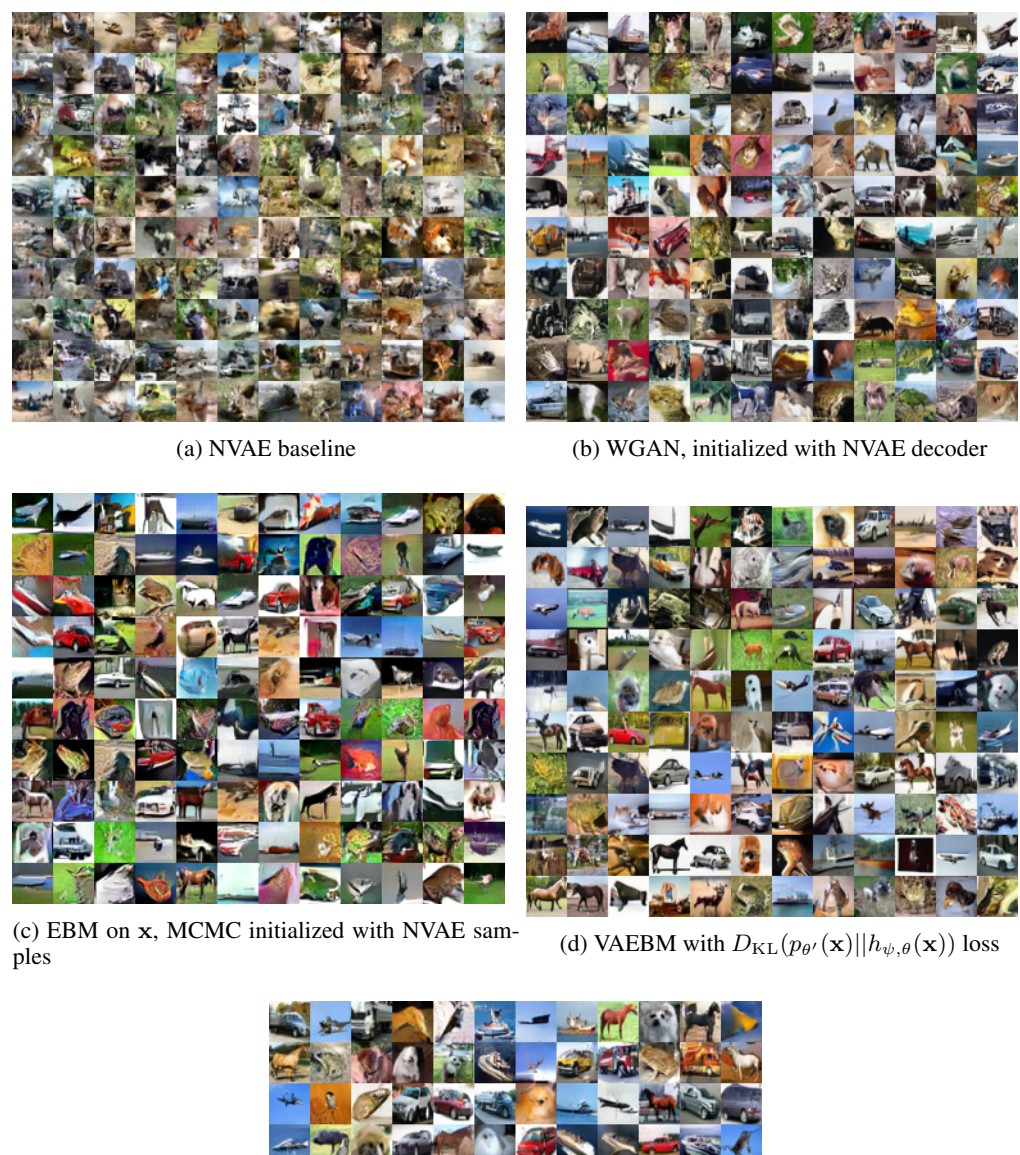

(a) NVAE baseline

(b) WGAN, initialized with NVAE decoder

(c) EBM on $\mathbf{x}$, MCMC initialized with NVAE samples

(d) VAEBM with $D_{\mathrm{KL}}(p_{\theta'}(\mathbf{x})||h_{\psi,\theta}(\mathbf{x}))$ loss

(e) VAEBM

Figure 7: Qualitative results of ablation study in Sec. 5.2. and Appendix C

before and after MCMC in Figure 15. Note that the samples used to visualize MCMC are generated by initializing MCMC chains with VAE samples with full temperature 1.0.

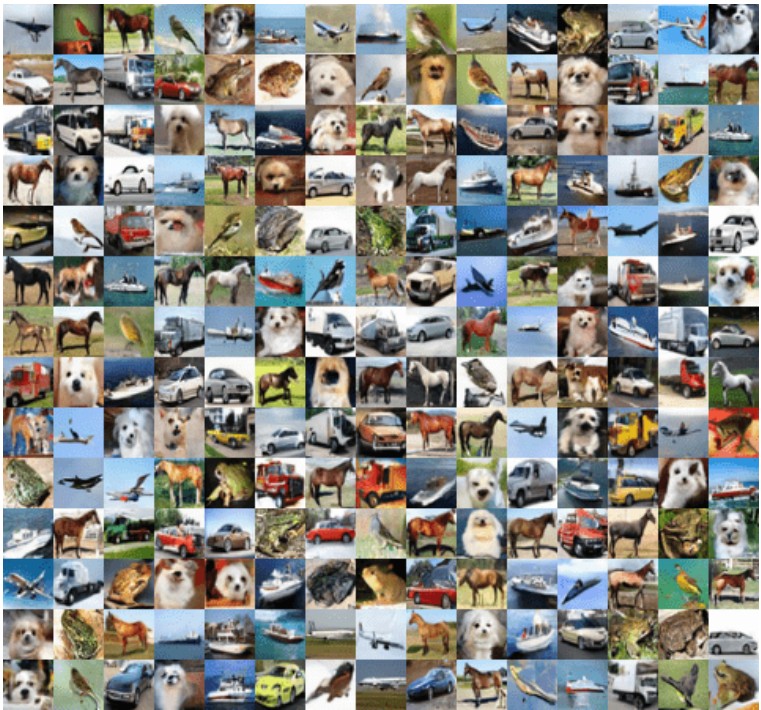

Figure 8: Additional CIFAR-10 samples

## I  NEAREST NEIGHBORS

We show nearest neighbors in the training set with generated samples on CIFAR-10 (in Figure 16 and 17) and CelebA HQ 256 (in Figure 18 and 19). We observe that the nearest neighbors are significantly different from the samples, suggesting that our models generalize well.

## J  SETTINGS OF SAMPLING SPEED EXPERIMENT

We use the official implementation and checkpoints of NCSN at `https://github.com/ermongroup/ncsn`. We run the experiments on a computer with a Titan RTX GPU. We use PyTorch 1.5.0 and CUDA 10.2.

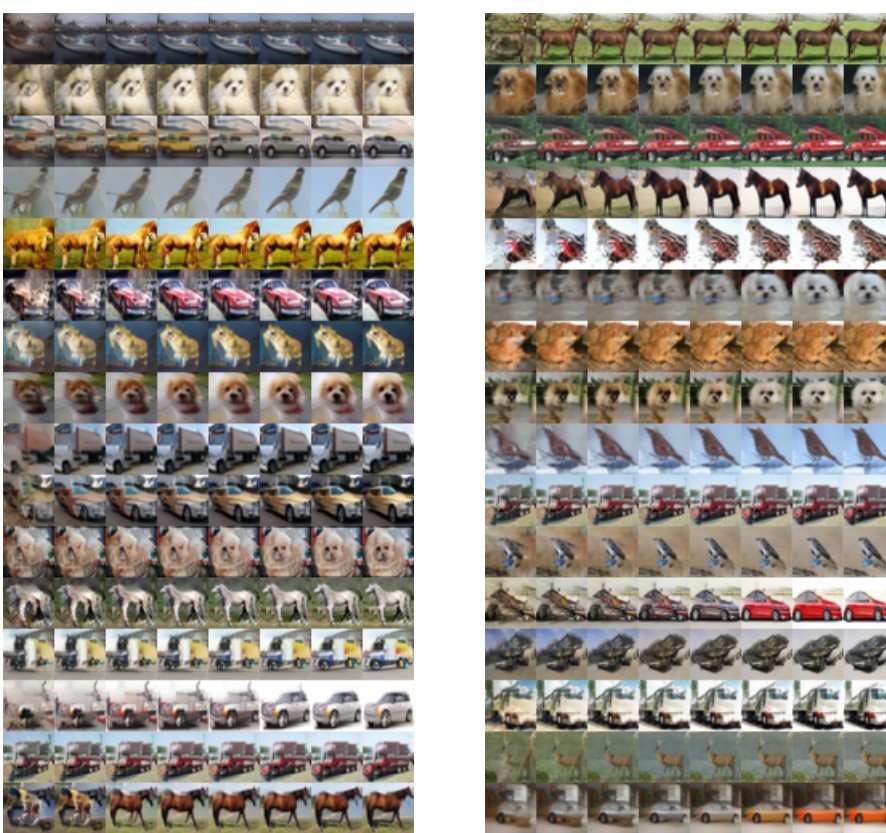

Figure 9: Additional visualizations of MCMC chains when sampling from the model for CIFAR-10

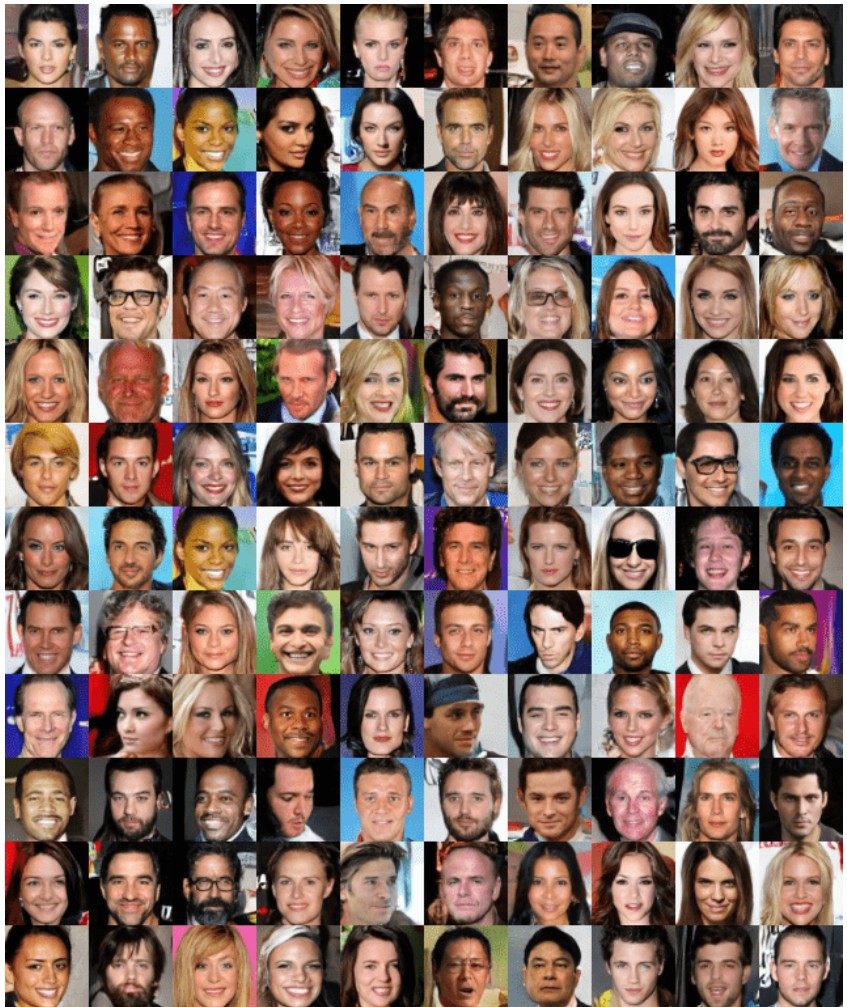

Figure 10: Additional CelebA 64 samples

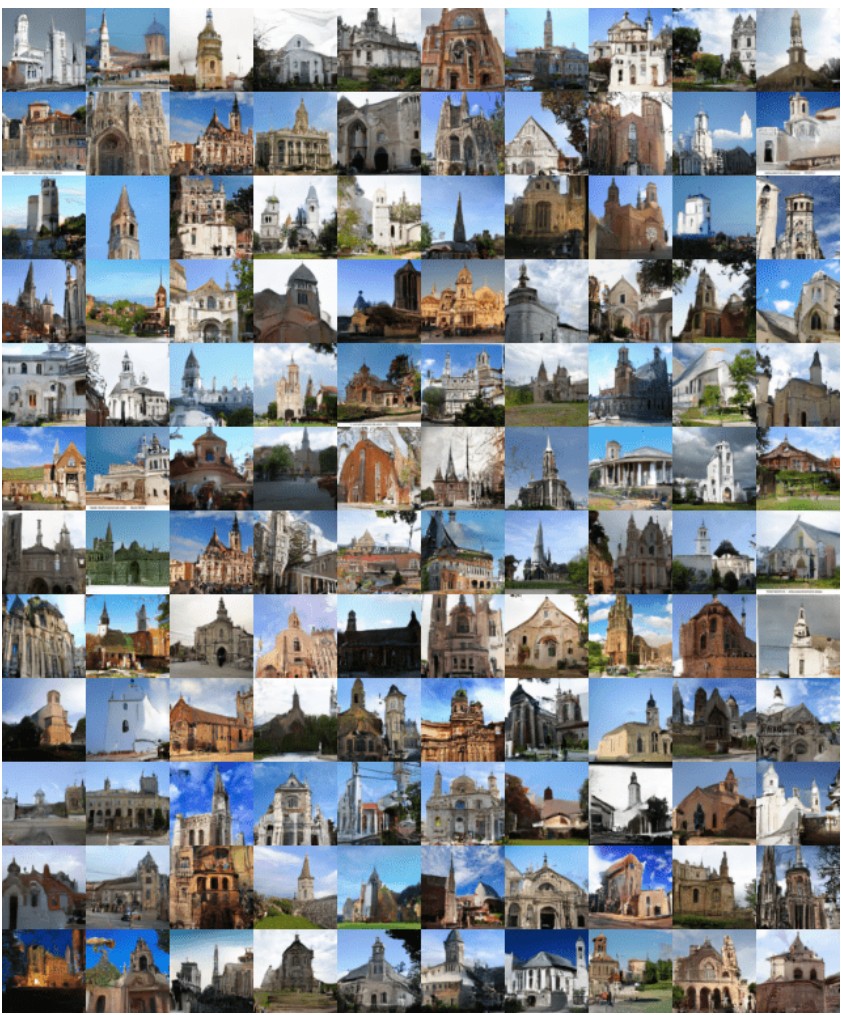

Figure 11: Additional LSUN Church 64 samples

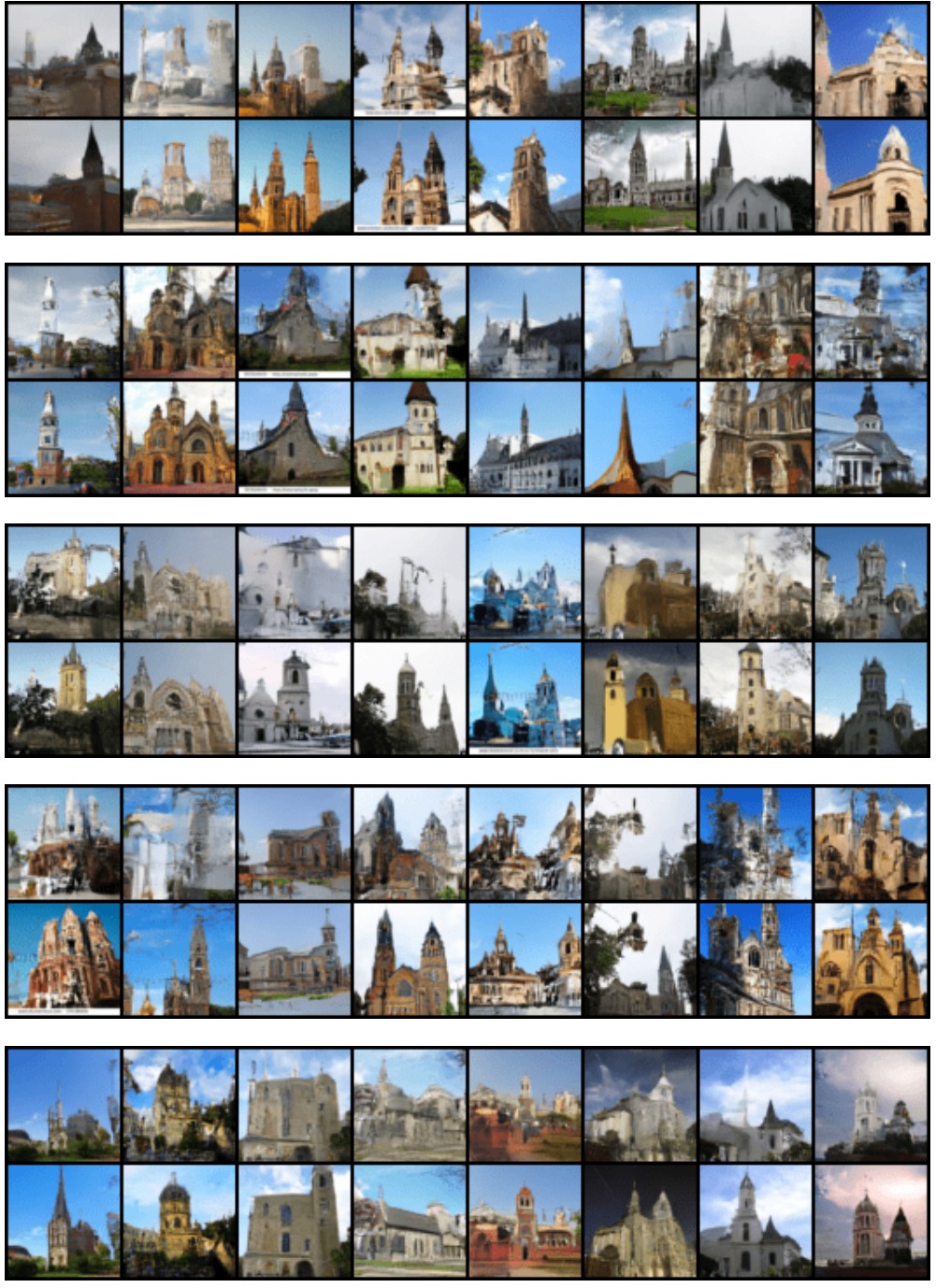

Figure 12: Visualizing the effect of MCMC sampling on LSUN Church 64 dataset. For each subfigure, the top row contains initial samples from the VAE, and the bottom row contains corresponding samples after MCMC. We observe that MCMC sampling fixes the corrupted initial samples and refines the details.

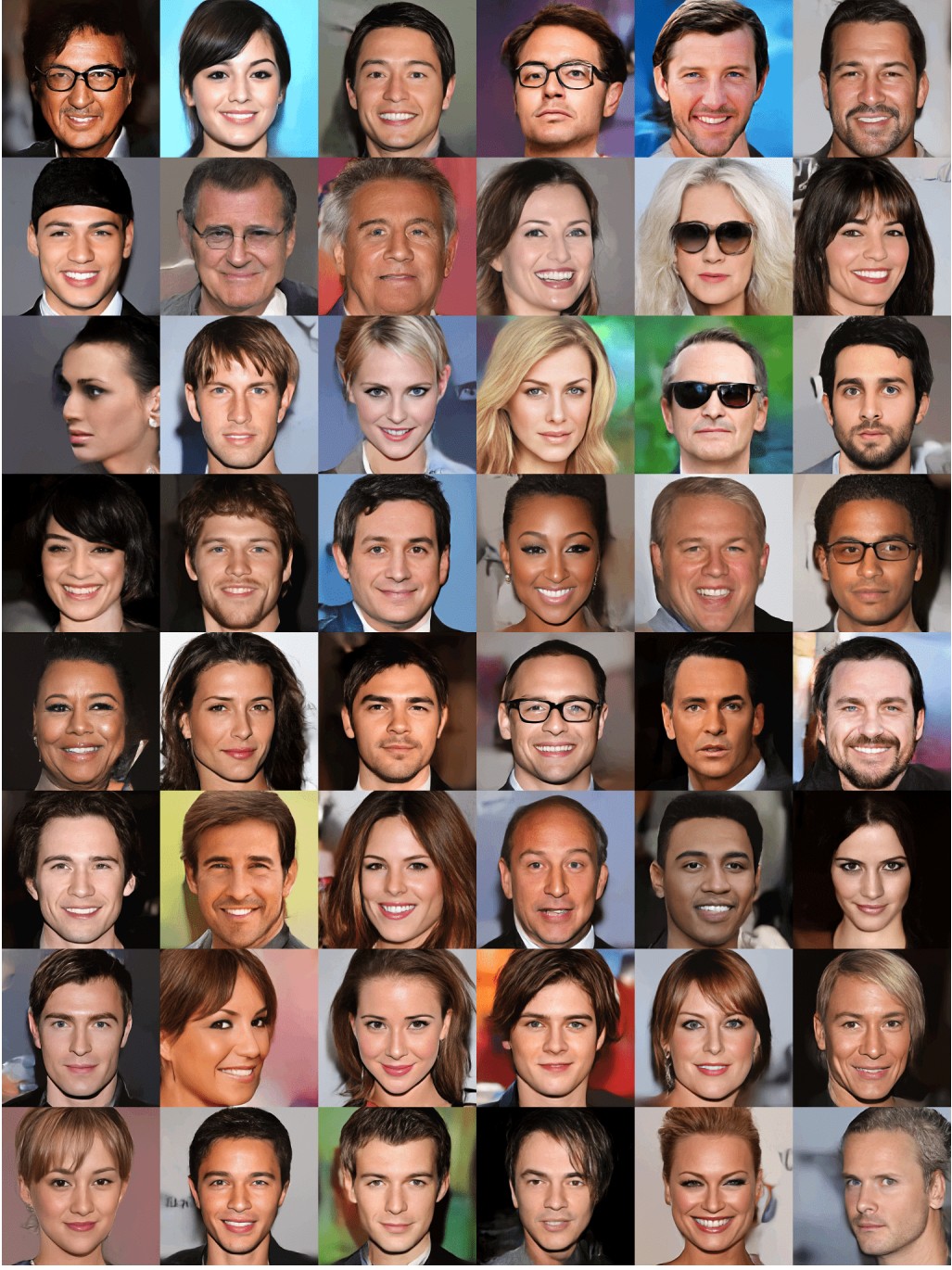

Figure 13: Additional CelebA HQ 256 samples. Initial samples from VAE for MCMC initializations are generated with temperature 0.7. Samples are uncurated.

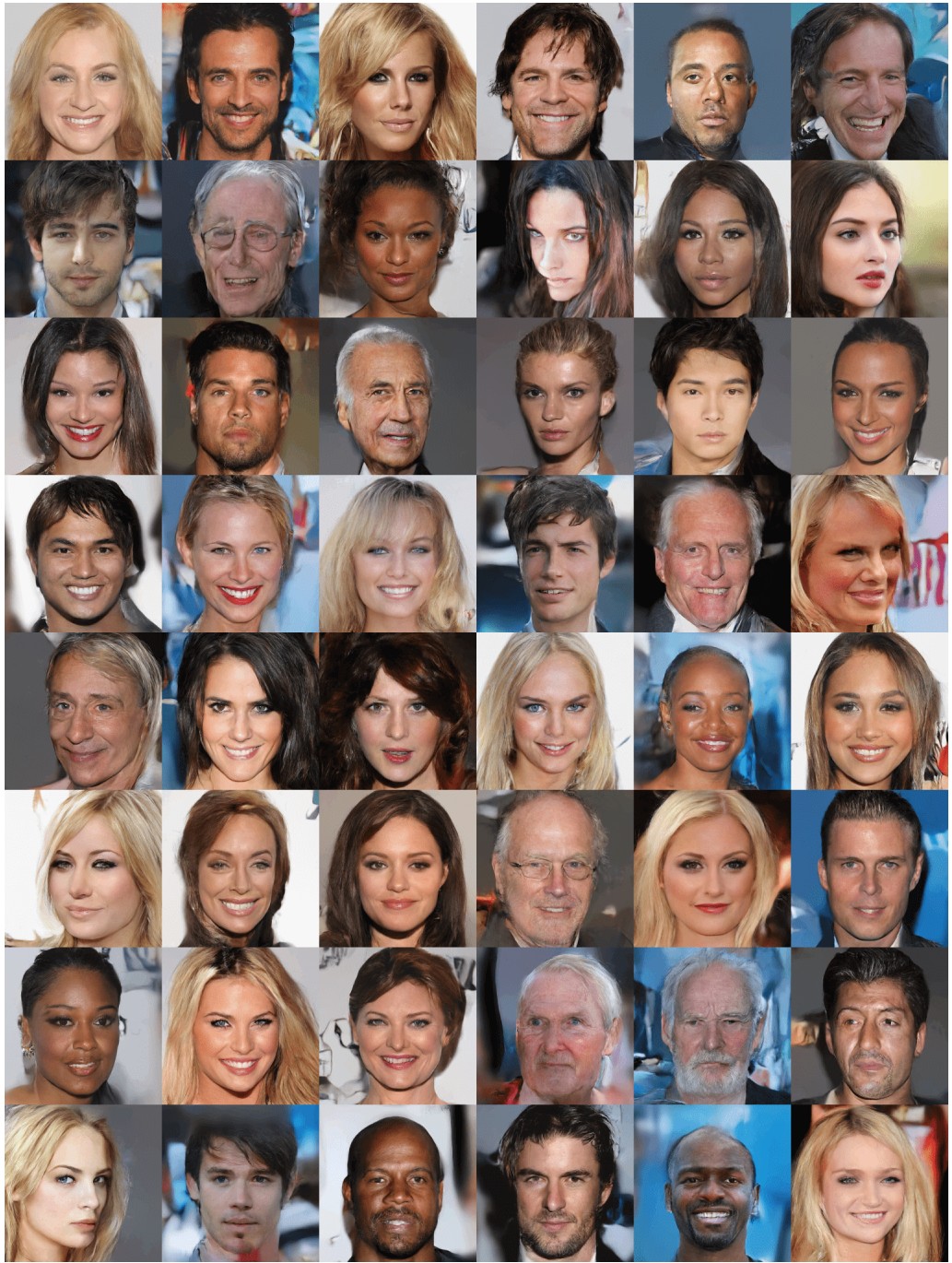

Figure 14: Additional CelebA HQ 256 samples. Initial samples from VAE for MCMC initializations are generated with full temperature 1.0. Samples are uncurated.

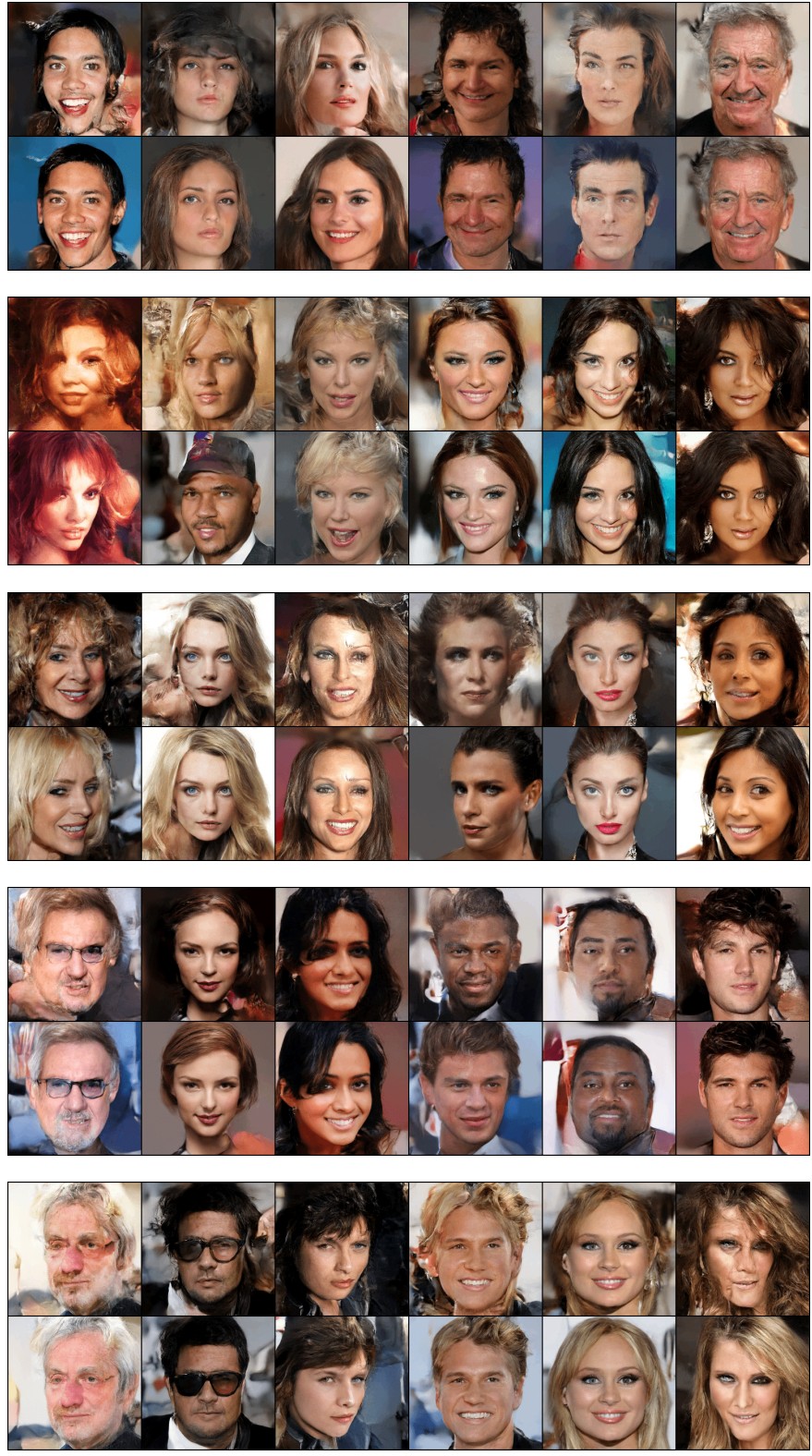

Figure 15: Visualizing the effect of MCMC sampling on CelebA HQ 256 dataset. Samples are generated by initializing MCMC with full temperature VAE samples. MCMC sampling fixes the artifacts of VAE samples, especially on hairs.

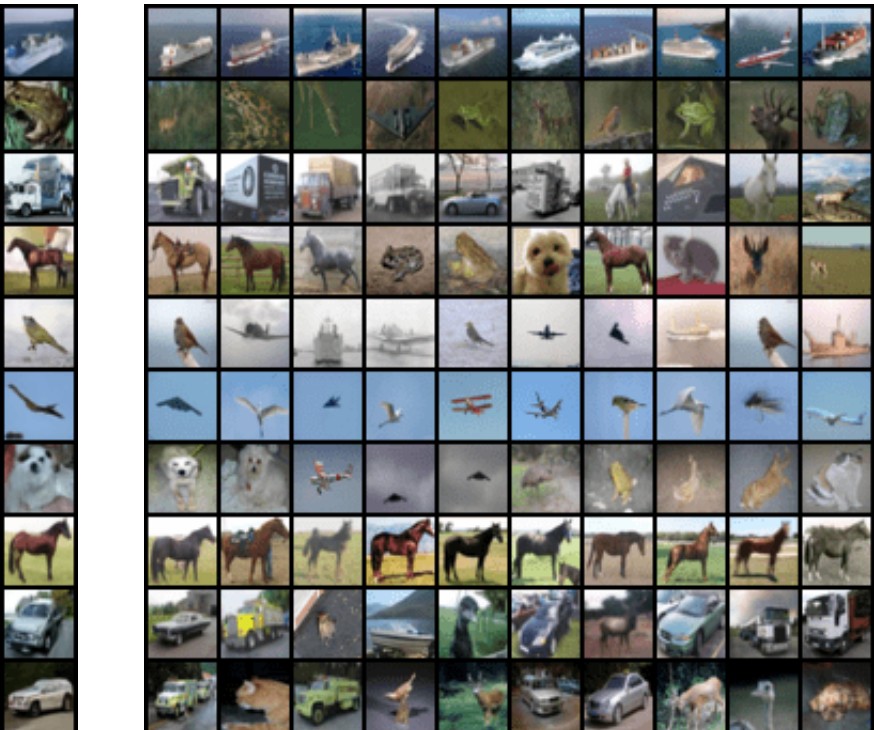

Figure 16: CIFAR-10 nearest neighbors in pixel distance. Generated samples are in the leftmost column, and training set nearest neighbors are in the remaining columns.

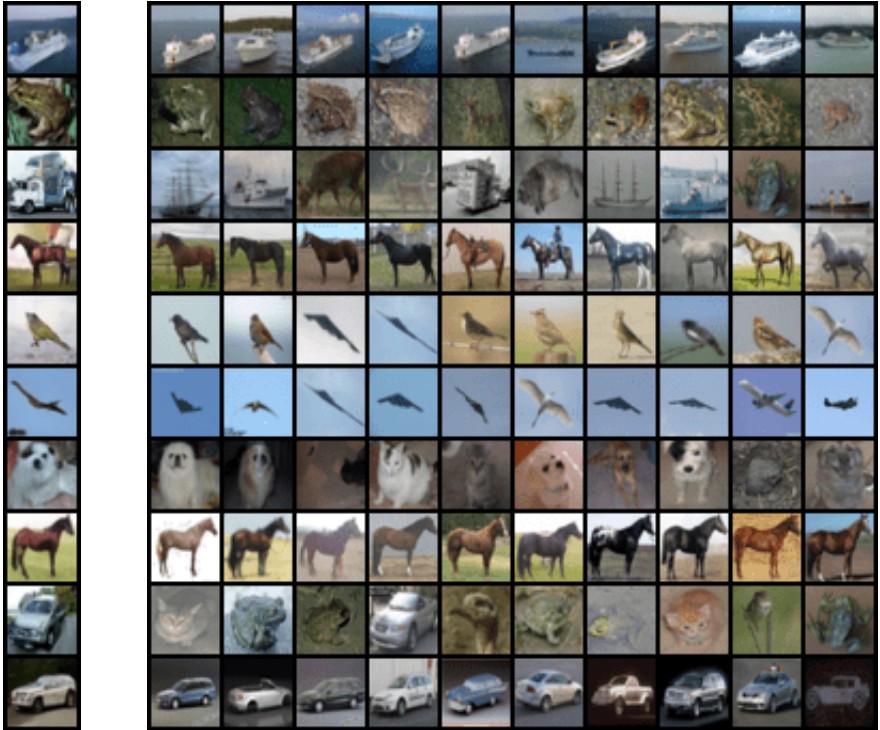

Figure 17: CIFAR-10 nearest neighbors in Inception feature distance. Generated samples are in the leftmost column, and training set nearest neighbors are in the remaining columns.

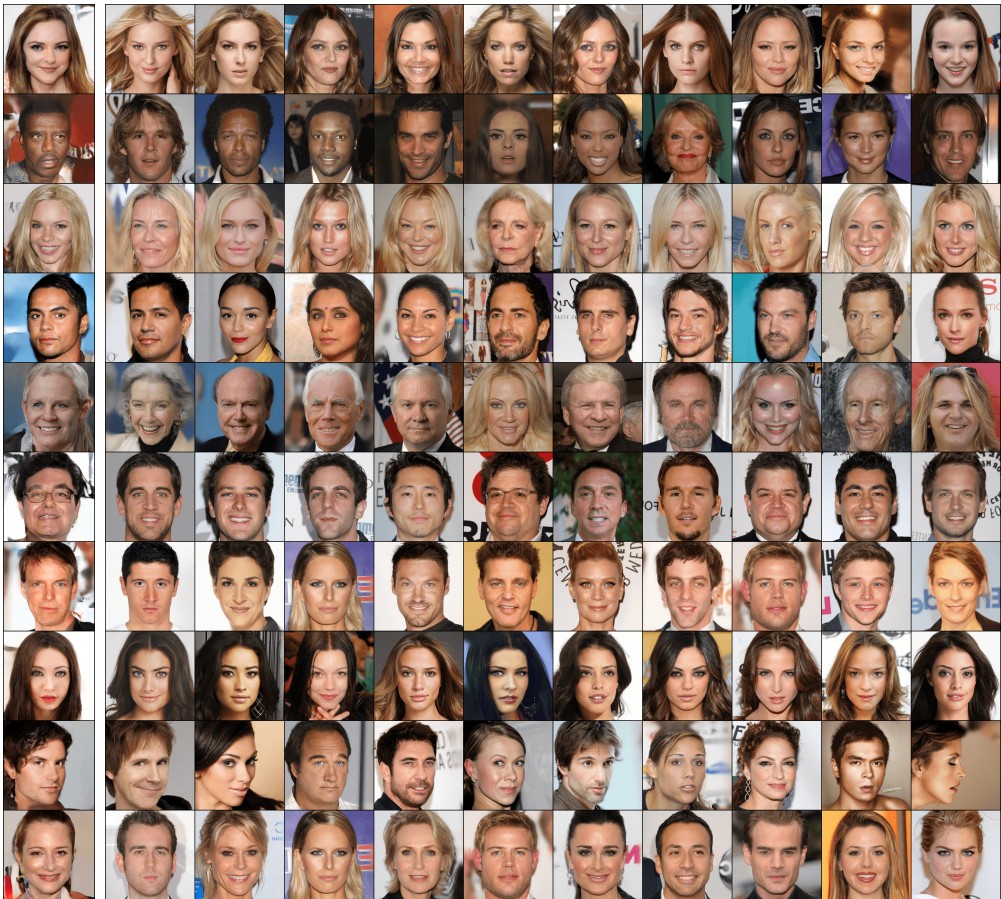

Figure 18: CelebA HQ 256 nearest neighbors in pixel distance, computed on a $160 \times 160$ center crop to focus more on faces rather than backgrounds. Generated samples are in the leftmost column, and training set nearest neighbors are in the remaining columns.

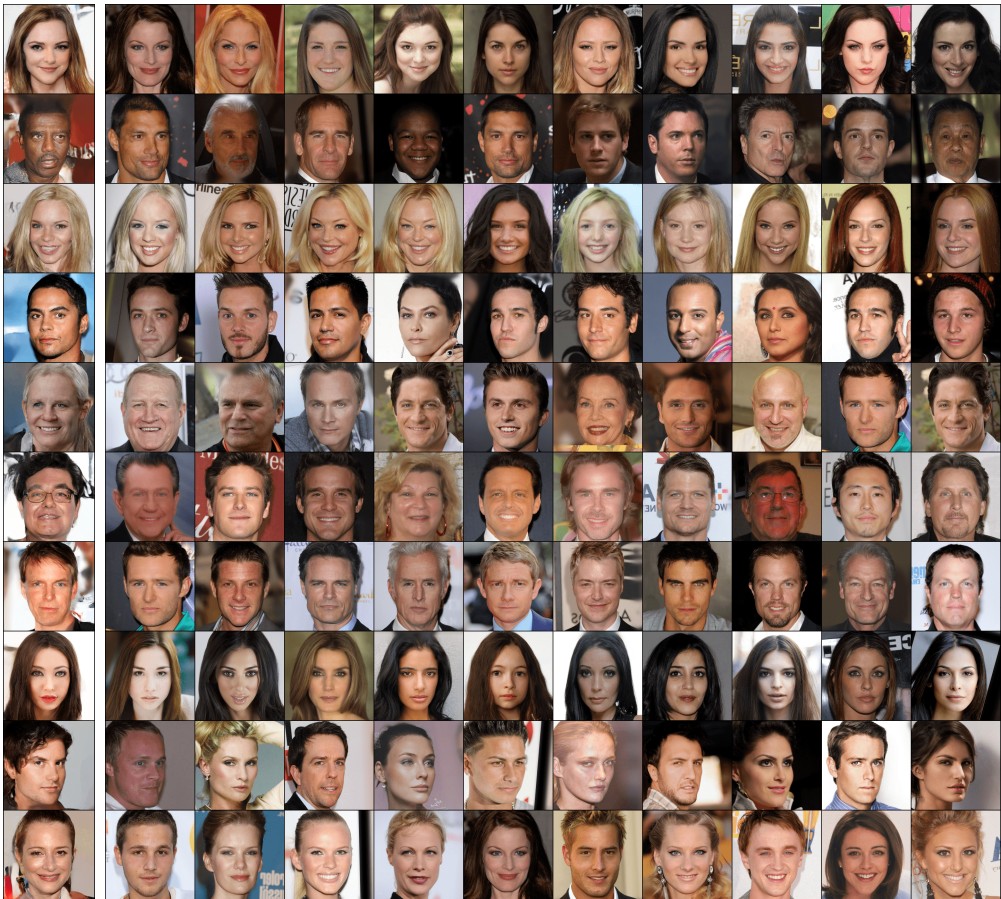

Figure 19: CelebA HQ 256 nearest neighbors in Inception feature distance, computed on a $160 \times 160$ center crop. Generated samples are in the leftmost column, and training set nearest neighbors are in the remaining columns.

