# OpenReview forum: "VAEBM: A Symbiosis between Variational Autoencoders and Energy-based Models"
_ICLR.cc/2021/Conference — ICLR 2021 Spotlight_

### Official Review · AnonReviewer4 · 2020-10-27
**A promising symbiosis between VAEs and EBMs**

**Rating:** 8
**Confidence:** 4

**Review:**

The authors propose a generative model that is a combination (product) of a VAE and an EBM, where the goal of the EBM is to reduce the probability of out-of-manifold samples, which are typically generated by VAEs. The authors propose efficient training and sampling procedures, in which the VAE is trained first and during the EBM negative-phase, samples are drawn from the joint (x, z) VAE space using reparameterization. The method is shown to achieve high quality samples on several modern image datasets, good FID scores and mode coverage. Ablation studies show the contribution of the different elements.

This is, in my opinion, a very good work, which combines a novel and well-motivated idea with clear writing and extensive experimental evidence.

Some comments and questions:
- Does the separate twos-stage training enable the model to reach the optimal point that can be reached in joint training, or is it an approximation? If its an approximation, I think it should be discussed or perhaps bounded.
- Does the combined model allow computing the likelihood? Can it be evaluated and compared to other models in terms of bits/dimension (e.g. as in VAE or NVAE)?
- It might be interesting (not something that I think is mandatory) to measure the NVAE log-likelihood of samples generated by the combined model compared to samples generated just by the NVAE.

To summarize:
pros:
- novelty
- significance
- experimental evidence
- quality of writing

cons:
- combining two separately trained models - perhaps sub-optimal

**Update: I thanks the authors for their answers and revised version and keep my positive rating.**

---

> ### Author Response · Authors · 2020-11-16
> **Response to Reviewer 4**
>
> We thank the reviewer for providing positive feedback. We address the reviewer’s comments and concerns below.
>
> * Regarding the two-stage training
>
> The two-stage training shares some similarities with coordinate descent algorithms, where the parameters are split into two parts and we update one part of parameters while keeping the other part fixed. We are not optimizing an approximate objective; instead, we are just using an alternative optimization approach. The two-stage optimization may lead to suboptimal solutions (actually we cannot easily guarantee optimal solutions with any optimization algorithm); however, we enjoy huge benefits brought by two-stage training. Per-step updates for the energy network are very expensive due to the MCMC, and pre-training the VAE provides a good initial approximation to the data density, so the energy network that corrects and modifies the density can be trained with a small number of updates (only a few epochs, as stated in Appendix E).
>
> * Exact likelihood estimation
>
> Estimating the exact likelihood of EBMs is a challenging task. Typically, the estimation involves annealed importance sampling (AIS), as done in [1]. This is computationally intense, as [1] reported that it took over 2 days on CIFAR-10. Our model has a large VAE component, so the time needed is even longer. More importantly, AIS will provide a stochastic lower bound on $\log Z$, and since the log likelihood of EBM involves negative $\log Z$, it will overestimate the log likelihood. In contrast, estimating the likelihood of the VAE (for example, by computing the IWAE bound) returns a lower bound on log likelihood. This makes the log likelihood estimates for EBM and VAE not comparable (we cannot compare a lower bound with an upper bound).
>
> Alternatively, as suggested by reviewer 2, we conduct a likelihood estimation experiment on a 2-d toy dataset, where we can accurately estimate $\log Z$ by numerical integration. We use the commonly used 25 Gaussian Mixtures data, and we present this experiment in **section 5.4 in the revised draft**. Results show that our model obtains better log likelihood (-1.5 nats) than the VAE (-2.9 nats), and it is close to the likelihood under the true distribution (-1.1 nats). Please refer to section 5.4 for more details.
>
> Thank you again and we will be very happy to answer any of your concerns or questions in the remaining rebuttal period.

---

### Official Review · AnonReviewer1 · 2020-10-28

**Rating:** 6
**Confidence:** 3

**Review:**

Pros:
this method proposed to use VAE+EBM for generative modelling. Unlike other VAE+GAN/EBM-liked model, it added a EBM after VAE.
Overall method is easy to understand and follow.
To accelerate the training, the authors also applied a buffer to store the previous examples for easy sampling.

Cons:
In the experiment, the authors compared other models with VAEBM, it is reasonable to compare the results with reported scores in other works, however, since the architecture is a fairly important factor (such that swish instead relu, resblock instead of cnn, weight norm instead of spectral norm), etc, is it also reasonable that the improvement is partially contributed by such design of architecture.
So I will suggest that the authors should use the same architecture design (choose other one or two models for all tasks), and test whether the proposed method can actually gain that much of improvement.

---

> ### Author Response · Authors · 2020-11-16
> **Response to Reviewer 1**
>
> We would like to thank the reviewer for providing positive feedback. We address the concerns regarding the discrepancy between our energy networks and the networks used in [1].
>
> On CIFAR10 where we compare our results with [1], we use the energy network with exactly the same backbone structure (i.e, number of residual blocks, number of channels in each block) as [1]. In particular, [1] also uses resblocks and therefore we do not replace CNN with resblocks. This suggests that our networks have the same number of parameters as those in [1].
>
> [1] reported results using LeakyReLU activations. However,  in their Appendix A.12, they said “we found that using either ReLU, LeakyReLU, or Swish activation in EBMs lead to good performance. The Swish activation in particular adds a noticeable boost to training stability.” Our observation is similar: in our model, LeakyReLU and Swish both work well, and Swish is slightly better in terms of performance and stability. Therefore we use Swish throughout the experiments.
>
> We indeed find that replacing spectral normalization (SN) with weight normalization (WN) leads to a performance boost. However, this is not applicable to the training of EBMs on data space as in [1]. We tried to train a EBM on data space without SN and with WN, but the training quickly diverged. We conjecture that training VAEBM is more stable than training EBM on data space, and therefore we can drop some strong regularizations such as SN or the 2-norm regularization for the energy that significantly constrain the expressivity of the model. This can be another potential benefit of our method.
>
> In addition, as shown in table 4, we train an EBM on data space. The FID score is similar to the one reported in [1], which is significantly worse than our model. We want to emphasize that we have tried several different settings to improve this baseline, and therefore we are confident that the large improvement of VAEBM over EBM on data space is not due to network architecture design.
>
> Thank you again and we will be very happy to answer any of your concerns or questions in the remaining rebuttal period.
>
> [1] Implicit Generation and Modeling with Energy-Based Models. Yilun Du and Igor Mordatch. https://arxiv.org/abs/1903.08689

---

### Official Review · AnonReviewer2 · 2020-10-29

**Rating:** 7
**Confidence:** 4

**Review:**

Strengths:
The paper provides a thorough overview of recent work towards training EBMs.
The approach generates high quality image samples by combining EBMs and VAE based models.
The paper is well written and is easy to follow
I find it quite interesting that a combination of both models leads to significant overall improved generative performance
I also enjoyed the proposed change in the paper -- and it seems to elegantly solve several problem in EBM training.

Weaknesses:
My most major concern is that since we are utilizing a maximum likelihood objective to train models, it would be good to evaluate  the overall likelihood of the trained model, even if only in the  2D domain.
The histogram of likelihoods of data points is a bit disappointing -- it falls a similar trend of other EBM models, but it would nicer if it followed a Gaussian distribution
What happens when more Langevin sampling steps are applied to the model? (greater than the few used in training)
I'm also curious on what sampling only the trained energy model looks like (without using the trained VAE parameterization) at evaluation time
I would also be curious to see how the trained EBM, with the VAE generator  can compose together with other models. See for example [1].

[1] Yilun Du, Shuang Li, Igor Mordatch. Compositional Visual Generation and Inference with Energy Based Models. NeurIPS 2020

#### Post Rebuttal-Update

I thank the authors for responding to my concerns. I enjoyed reading the paper and maintain my rating.

---

> ### Author Response · Authors · 2020-11-16
> **Response to Reviewer 2**
>
> We thank the reviewer for providing positive comments on our manuscript. Below, we address the concerns in detail.
>
> * Evaluating the overall likelihood of the trained model
>
> We thank the reviewer for suggesting likelihood estimation on a 2-d toy distribution. We think it is a very good idea, as we can estimate the partition function by numerical integration. In contrast, estimating normalized likelihood for EBMs is challenging when the data dimension is high. Techniques like annealed importance sampling for estimating the partition function are computationally expensive and inaccurate. Therefore, we run a simple likelihood estimation experiment on the widely used 25 Gaussian Mixtures data set, and we present this experiment in **section 5.4 in the revised draft**. Results show that our model obtains better log likelihood (-1.5 nats) than the VAE (-2.9 nats), and it is close to the likelihood under the true distribution (-1.1 nats). Please refer to section 5.4 of our revised draft for more details.
>
> * The histogram of likelihoods of data points
>
> We plot the histogram of likelihoods to show that the likelihoods of train and test data are similar, suggesting that our model generalizes well and does not overfit the training data.
>
> * More Langevin sampling steps are applied to the model
>
> When we examine long-run MCMC chains on real datasets, we observe that most chains still stay around the local mode. Other work such as [1] and [2] also reports that models trained with short-run MCMC have non-mixing long-run chains. For example, figure 2 in [2] shows that the long-run chains do not exhibit mode traversal regardless of whether the LD in training is convergent or not. We believe that in order to improve mixing further, we need better sampling techniques such as Hamiltonian Monte Carlo (HMC) for both training and test sampling. This would be an interesting direction for future work.
>
> * Sampling from only the trained energy model without VAE
>
> We think directly sampling from the energy model will not produce good samples.  In our model, the energy-based component is used to correct and modify the density of the distribution learned by the VAE. It can be thought of as learning the residual between the VAE’s distribution and true data distribution. Therefore, the energy function itself does not represent a meaningful distribution on x and we cannot sample from it without the VAE.
>
> * Composing our models with other models
>
> We thank the reviewer for the suggestion. Indeed, one of the main advantages of EBMs is their flexibility to compose with other models. We believe it is an interesting future direction to explore the composition of our model with other models for different tasks.
>
> Thank you again and we will be very happy to answer any of your concerns or questions in the remaining rebuttal period.
>
> [1] On learning non-convergent short-run mcmc toward energy-based model. Nijkamp et al.  https://arxiv.org/abs/1904.09770
>
> [2] On the Anatomy of MCMC-Based Maximum Likelihood Learning of Energy-Based Models Nijkamp et al. https://arxiv.org/abs/1903.12370

---

### Official Review · AnonReviewer3 · 2020-10-30
**Interesting paper that learns EBM as a correction of VAE.**

**Rating:** 7
**Confidence:** 5

**Review:**

This paper proposes a model that corrects VAE by an energy-based model defined on image space. The model is learned in two phase. The first phase learns the VAE model, while the second phase learns the EBM correction term by MLE. Experimental results show that the proposed method outperforms pure EBM defined on image space and also pure VAE models by large margins.

- pros: the paper is clear written and easy to follow. The ablation study shows clearly the advantage over baseline methods. Sampling from EBM on image space is hard. With VAE as a backbone, the sampling can be transferred to the latent space and the residual \epsilon in the image space, which is much more friendly to MCMC sampling.

- cons:
1. The energy term is used to correct only on image space. Would be interesting to see if VAE can be corrected by a latent EBM where the energy function is defined on (x, z).
2. After learning, would long-run MCMC sampling chain remain stable and mix well? It would be interesting to diagnose the long run chain behavior, and compare the difference of sampling in the space (\epsilon_x, \epsilon_z) and (x, z).
3. For the synthesized results of CIFAR-10, it seems that some patterns appear repeatedly (e.g., the white dog face). Is the model suffered from mode collapsing problem?

Overall, it is a good submission that proposes a principled method to combine VAE and EBM and demonstrates strong empirical results. I tend to accept this paper.

---

> ### Author Response · Authors · 2020-11-16
> **Response to Reviewer 3**
>
> We thank the reviewer for providing positive comments on our manuscript. Below we reply to each comment in detail.
>
> * Latent EBM where the energy function is defined on $(\mathbf{x}, \mathbf{z})$
>
> Thank you for the suggestion. We agree that modeling the energy in the joint $(\mathbf{x}, \mathbf{z})$-space is an excellent future extension of our current model. At this point, we choose to let the energy function only take x as input mainly for the motivation that we want to correct the distribution on data space learned by the VAE. We also borrow the motivation from GANs, where the discriminator network operates only on x.
>
> * Long-run MCMC sampling chain
>
> When we examine long-run MCMC chains on real datasets, we observe that most chains still stay around the local mode. Other works such as [1] and [2] also report that models trained with short-run MCMC have non-mixing long-run chains. For example, figure 2 in [4] shows that the long-run chains do not exhibit mode traversal regardless of whether the LD in training is convergent or not. We believe that in order to further improve mixing, we need better sampling techniques such as Hamiltonian Monte Carlo (HMC) for both training and test sampling. This would be an interesting direction for future work.
>
> * Compare with sampling in the $(\mathbf{x}, \mathbf{z})$-space
>
> Thanks for suggesting this experiment. This indeed can show the effectiveness of sampling in the reparameterized space. In the **newly added Appendix B.1**, we first show that sampling in the $(\mathbf{x}, \mathbf{z})$-space is mathematically equivalent to sampling in the $(\epsilon_x, \epsilon_z)$-space, if we adjust the LD step size by the variance of each component of $\mathbf{x}$ and $\mathbf{z}$. However, this is not easy to implement because in VAE’s reparametrization, each latent variable and pixel component has different variance. More importantly, our latent variable $\mathbf{z}$ in the prior follows block-wise auto-regressive Gaussian distributions, so the variance of each component $\mathbf{z_i}$ depends on the value of $\mathbf{z}_{<i}$. We foresee that because of this dependency, using a fixed step size per component of $\mathbf{z}$ will not be effective, even when it is set differently for each component. Working in $(\epsilon_x, \epsilon_z)$-space avoids such dependency. In this appendix, we empirically show that directly sampling from $(\mathbf{x}, \mathbf{z})$ without adjusting the steps size for each variable separately leads to poor sample quality. We hope this additional study demonstrates the benefits of sampling from the reparametrized distribution.
>
> * Possible mode dropping issues
>
> We thank the reviewer for pointing this out. It is an interesting observation that we did not notice before. We agree that the diversity among dog samples seems to be limited. Interestingly, we find such a phenomenon is also observed in other recent works such as [3][4] which obtains impressive FID scores. In [3]’s Figure 18 and [4]’s Figure 11, we observe that most of the generated dog images contain white front faces. One reason might be that the dataset is slightly biased towards this pattern. While our model tends to cover all the modes of the data distribution, as indicated by the quantitative results on StackedMNIST in table 5, overlapped train/test likelihood histogram in Figure 6 and the diversity of generated images on CelebA and LSUN, we admit that there might be imbalance on the weights assigned to the different modes. This imbalance may be more pronounced for challenging tasks such as modeling CIFAR10. On such a challenging dataset, unconditional generative models may not have enough capacity to model each mode exactly as the true distribution, and it is an interesting extension to use a conditional model so that modes within each class can be modeled more faithfully. Another possible explanation is that we use short-run MCMC to obtain samples, so theoretically speaking what we obtained are not exact samples from our model. The effect of imbalanced modes may be exacerbated by the inaccurate sampling (the sampling scheme may tend to always ignore some modes with lower density). We believe that this issue is an interesting direction for future research.
>
> Thank you again and we will be very happy to answer any of your concerns or questions in the remaining rebuttal period.
>
> [1] On learning non-convergent short-run mcmc toward energy-based model. Nijkamp et al. https://arxiv.org/abs/1904.09770
>
> [2] On the Anatomy of MCMC-Based Maximum Likelihood Learning of Energy-Based Models Nijkamp et al. https://arxiv.org/abs/1903.12370
>
> [3] Denoising Diffusion Probabilistic Models. Ho et al. https://arxiv.org/abs/2006.11239
>
> [4] Learning Energy-Based Models by Diffusion Recovery Likelihood https://openreview.net/forum?id=v_1Soh8QUNc

---

### Author Response · Authors · 2020-11-16
**Manuscript updated and responses submitted**

Dear reviewers,

We want to thank all reviewers for your useful comments. We have submitted responses to all official reviews and uploaded an updated draft. The new draft mainly adds two sections:

1. Section 5.4 which includes an experiment on exact likelihood estimation on 2-d toy dataset. Results show that VAEBM greatly improves the test data likelihood over the VAE.

2. Appendix B.1 that studies sampling in $(\mathbf{x}, \mathbf{z})$-space versus sampling in $(\mathbf{\epsilon}_x, \mathbf{\epsilon}_z)$-space. We show that sampling in the $(\mathbf{x}, \mathbf{z})$-space is mathematically equivalent to sampling in the $(\mathbf{\epsilon}_x, \mathbf{\epsilon}_z)$-space, if we adjust the LD step size by the variance of each component , and directly sampling from $(\mathbf{x}, \mathbf{z})$-space without adjusting the steps size for each variable separately leads to poor sample quality.

We hope our responses addressed your concerns, and we will be very happy to answer any of your concerns or questions in the remaining rebuttal period. Thank you!

Authors of submission 562

---

### Decision · Program_Chairs · 2021-01-07
**Final Decision**

**Decision:**

Accept (Spotlight)

**Comment:**

This work presents a method to combine EBMs and VAEs in two stages. First, the VAE model is learned; second, an EBM-based correction term is learned via MLE. The methodology is novel and of interest to the ICLR community.